# A Review on Mechanical Models for Cellular Media: Investigation on Material Characterization and Numerical Simulation

**DOI:** 10.3390/polym13193283

**Published:** 2021-09-26

**Authors:** Guoqiang Luo, Yuxuan Zhu, Ruizhi Zhang, Peng Cao, Qiwen Liu, Jian Zhang, Yi Sun, Huan Yuan, Wei Guo, Qiang Shen, Lianmeng Zhang

**Affiliations:** 1State Key Lab of Advanced Technology for Materials Synthesis and Processing, Wuhan University of Technology, Wuhan 430070, China; luogq@whut.edu.cn (G.L.); zhuyx_whut@163.com (Y.Z.); zhangjian178@whut.edu.cn (J.Z.); sunyiwhut@163.com (Y.S.); lmzhang@whut.edu.cn (L.Z.); 2National Key Laboratory of Shock Wave and Detonation Physics, Institute of Fluid Physics, China Academy of Engineering Physics, Mianyang 621900, China; zhangrz1991@gmail.com; 3College of Architecture and Civil Engineering, Beijing University of Technology, Beijing 100124, China; 4Hubei Key Lab of Theory and Application of Advanced Materials Mechanics, Department of Mechanics and Engineering Structure, Wuhan University of Technology, Wuhan 430070, China; qiwen_liu@whut.edu.cn; 5School of Automotive Engineering, Wuhan University of Technology, Wuhan 430070, China; phoebe456@163.com (H.Y.); whutgw@whut.edu.cn (W.G.)

**Keywords:** cellular medium, constitutive relationship, homogenization algorithm, finite element method, mechanical model

## Abstract

Cellular media materials are used for automobiles, aircrafts, energy-efficient buildings, transportation, and other fields due to their light weight, designability, and good impact resistance. To devise a buffer structure reasonably and avoid resource and economic loss, it is necessary to completely comprehend the constitutive relationship of the buffer structure. This paper introduces the progress on research of the mechanical properties characterization, constitutive equations, and numerical simulation of porous structures. Currently, various methods can be used to construct cellular media mechanical models including simplified phenomenological constitutive models, homogenization algorithm models, single cell models, and multi-cell models. This paper reviews current key mechanical models for cellular media, attempting to track their evolution from their inception to their latest development. These models are categorized in terms of their mechanical modeling methods. This paper focuses on the importance of constitutive relationships and microstructure models in studying mechanical properties and optimizing structural design. The key issues concerning this topic and future directions for research are also discussed.

## 1. Introduction

A cellular medium is a porous structure with a solid material (polymer/metal/ceramic) as the matrix and a large number of bubbles. It is a very common material in nature, especially in the biological field. Cellular media provide a variety of protective measures and structures for organisms, and have high specific strength (e.g., turtle shells and wood) [1,2,3,4,5,6,7]. Inspired by nature, researchers began to have a strong interest in the study of the diversity and mechanical properties of porous media, and began to design and analyze the mechanical behavior of different porous media materials and structures. Cellular media materials have been widely used in packaging boxes, transportation, for daily necessities, industrial manufacturing, national defense, military industries, and in biomechanical research (substrate material of elastic matrix sensor) [8], due to their many advantages including their efficient heat and sound insulation, good compression, mechanical properties and ductility, and low cost [9,10,11,12,13,14].The mechanical properties of a cellular medium are determined by both matrix and cell structure. Investigating the constitutive relationship between these can help us better understand the mechanical properties of a material and have an irreplaceable effect on the optimization of its properties [15,16,17].

The internal structure deformation and evolution of cellular media is complex, and there are many problems in traditional experimental methods. Therefore, numerical simulation based on fine analysis model has become a valid research method. There are two numerical simulation methods to investigate the mechanical properties of cellular medium. One method ignores the microstructure and explores the simplified constitutive relationship. The other method builds the relationship between the mechanical properties and microstructure. Then, the geometric modeling of the cellular medium becomes critical.

In this paper, we first introduce the characterization of a cellular medium’s mechanical properties. Then, we provide the latest comments of key mechanical models, including classification of these models based on modeling scale and methodology, as well as a discussion of their advantages and disadvantages. We summarize and analyze these models in detail in Table 1. Finally, we propose prospects for these mechanical models. This work mainly focuses on the research progress of cellular medium materials in recent years, provides a reference and a compilation of important documents for researchers in the same research direction, and points out the focus of the research and potential application directions.

## 2. Characterization of Cellular Medium Mechanical Properties

The research on cellular media initially studied porous materials in nature [1,18,19,20,21,22,23,24,25,26,27], such as grapefruit peels, bamboo, turtle shells, and bone, as shown in Figure 1. After a large number of artificial cellular media appeared, comprehensive and systematic research began in the 1950s. Artificial cellular media materials include foam materials obtained by foaming, two-dimensional (2D) honeycomb materials, and cellular materials with various structures obtained by three-dimensional (3D) printing, as shown in Figure 2a–c. Since 1980, Gibson and Ashby [28] had systematically investigated the mechanical behavior of a variety of porous materials starting from two-dimensional honeycomb materials. They made a unified description and analysis of the structure and properties of a variety of porous materials (i.e., polymer foam, metal foam, ceramic foam, wood, and cancellous bone).

A cellular medium has its own unique characteristics. Compared with thin-walled tube and laminate composite materials, cellular media materials are isotropic and have a relatively large compressive strain, stable stress plateau, and lower density. The void porosity can exceed 90%, and they can be made into ultralight components. It is precisely due to the fact that cellular media materials have these excellent properties that they have received increasing attention in material research.

Previous research on cellular media has mainly focused on the influence of structural unit morphology, relative density, cell size, and defects on the material modulus, yield strength, and plateau stress. Research has also been conducted on the cell deformation characteristics of cellular media under quasi-static and dynamic loading conditions. The schematic diagrams of quasi-static loading and dynamic loading are shown in Figure 3. However, research on the energy absorption characteristics of cellular media under dynamic loads, especially the response characteristics under explosive loads, has not been fully investigated.

### 2.1. Quasi-Static Mechanical Properties

The mechanical properties of cellular media are significantly influenced by relative density, ambient temperature, and compressive strain rate. The stress-strain curve can be divided into three stages when the cellular medium is compressed, namely linear elasticity, plateau, and densification, as shown in Figure 2d. When the strain is small, the stress-strain curve shows linear elasticity, and the stress causes the cell struts to bend. Then, a plateau appears, and the stress is almost constant as the strain increases. The cell walls and struts yield, corresponding to the plastic deformation of the cellular medium. Finally, as the cell walls are squeezed together, the cellular medium is compacted, and the stress increases rapidly. The mechanical properties of cellular media are affected by their matrix properties, relative density (void porosity), and cell size, as shown in detail in Table 2.

Some researchers found that under the same porosity, the mechanical properties did not change with the cell size [35,36,37,49,50]. Some researchers hold different views. Smits [38] found that when the cell size was large, the compressive strength of PU foam decreased as the cell diameter decreased. However, when the cell diameter decreased to a certain extent, the compressive strength increased as the cell diameter decreased. The increase in compressive strength came from the increase in the polymer fraction in the cell walls [51,52,53]. Chen [39] investigated the mechanical behavior of 3D layered nanofoam, and believed that the solid surface had fewer atoms than the inside (so it had excess surface energy). They found as the cell cross-sectional diameter decreased, the influence of surface effects on modulus and strength increased. Notario [40] prepared PMMA foams with porosity of 0.5 and cell sizes of 200 nm to 11 μm. Compared with microcellular foams, the thermomechanical properties and Shore hardness of nanocellular foams were significantly improved by 25% and 15%. Zhu [30] investigated the effect of void porosity, cell size, and cell shape on the static-compressive properties of PMMA foams. They found that porosity had greatest influence on the mechanical properties [54], cell size and cell shape also affect the compressive properties.

According to the above examples from the literature, cell sizes have different effects on the mechanical properties of cellular medium. We agree with Smits [38] that while the cell size has a turning point, different relative densities and matrix materials correspond to different turning points. Therefore, the correlation between mechanical properties and cell structure requires further research.

### 2.2. Dynamic Mechanical Properties

As a cushioning material, cellular media are subjected to dynamic loads in many practical applications, and the loading process is usually accompanied by a high loading strain rate. The protective foam in the side door panels of automobiles has a strain rate of up to 1500 s^−1^ when the velocity of side collision is 90 km/h. The strain rate of the foam energy-absorbing box of the automobile reaches 200 s^−1^ at 40 km/h [47]. When the cellular medium is subjected to larger strain and higher strain rate under dynamic loading, the static test result and model cannot exactly evaluate the mechanical properties of the cellular medium. Therefore, for high strain rate loading scenarios, it is important to investigate their mechanical behavior under dynamic loads [34,55,56,57].

There are different opinions on whether cellular media have strain rate effects. Some researchers found that the strain rate had little effect on cellular medium under dynamic compression [41,42,43,58]. In addition, some studies have shown that cellular media have a strong strain rate effect [31,44,45,46,47,48]. The deformation modes of cellular media under dynamic load are significantly different from that under quasi-static load. Cellular media have obvious deformation localization and stress enhancement under dynamic load [59,60,61,62].

Ramachandra [45] used the pressure head test to find that the effect of energy absorption on closed-cell foams with a relative density of 8% at a velocity below 10 m/s basically did not change, but at a velocity above 10 m/s, its energy absorption effect increased significantly. Sun [46] proposed that the strain rate effect of cellular media was closely related to the matrix material and cell structure. Different microstructures of cellular media experience different deformation modes and stress and strain distributions during the compression. Bouix [47] investigated the stress-strain behavior of PP foams in the strain rate range of 0.01–1500 s^−1^ and found that low-porosity foams had strain rate hardening due to thicker cell walls, resulting in high yield stress. However, the high-porosity foams were not sensitive to strain rate hardening. Luong [31] found that the yield strength of PVC foam material increased with the increase of strain-rate, as shown in Figure 2e. Chen [48] found that when the strain-rate exceeded 113 s^−1^, the compressive strength of EPS foam increased rapidly with the increase of strain-rate, and the densification strain decreased slightly with the increase of strain-rate. In the strain-rate range of 1–280 s^−1^, Young’s modulus had no obvious strain rate dependence.

There are also different opinions on whether cellular media can truly attenuate stress waves. Researchers used shock wave generators, explosives, and light gas guns to load metal foams to study the attenuation effect of shock waves [63]. They found that foams had better attenuation characteristics for shock waves, and the shock wave pressure decayed exponentially. Cruttmann [64] used finite element simulations to find that cellular media mainly absorbed impact loads through internal cell deformation and collapse to attenuate stress waves. Branch [65,66] found that the periodically arranged cell structure [67] can adjust the elastoplasticity of a cellular medium. Cellular media materials with simple cubic structures were prone to stress concentration, which induced emission flow and attenuated impact load energy. Cellular media materials with face-centered cubic structures were not prone to stress concentration (due to layer-to-layer symmetry) and did not induce emission flow. Radford [68] used aluminum foam as a bullet to impact a dense material, and investigated the propagation of shock waves in aluminum foam. They found that the energy carried by the shock wave was severely dissipated in the deformation of the cell structure, and the shock wave stress was related to the density of the aluminum foam. 

Most of the research on the dynamic mechanical properties of cellular media are concentrated in the range of low and medium strain rates. We believe that there is now a turning point in cellular media. Above this point it has a strain rate effect, and the turning point corresponding to different relative density, cell size, and matrix material is different. Regarding the high strain rate loading of cell media, especially the strain rate higher than 10^5^ s^−1^ and the mechanical response behavior under explosive loading, it is necessary to thoroughly study the complex wave system effects inside.

## 3. The Constitutive Relationship of Cellular Medium

The constitutive relationship can be written as the evolution equation of generalized force and generalized displacement with time. For the mechanical properties of solid materials, the constitutive relationship can be expressed as a stress-strain curve. The constitutive relationship of a cellular medium is greatly affected by relative density, temperature, and strain rate. A macroscopic theoretical cellular medium model can be constructed by experimentation to characterize the foam cushioning performance under specific conditions. Therefore, knowledge of mechanical properties is a prerequisite for establishing the constitutive relationship.

Through investigating of constitutive equations, we can fully understand the mechanical properties of materials and optimize their material properties. Many researchers put forward simplified constitutive models, which can be import into the finite element model to facilitate the calculation of complex mechanical properties [55,69,70,71,72,73,74,75,76].

### 3.1. Phenomenological Constitutive Model

The phenomenological constitutive model is the mathematical expression between stress and strain obtained by fitting experimental data. It can be used to describe physical phenomena such as linear elastic region, plateau region, and densification region. However, it cannot explain the microstructure deformation and failure mechanism of the foam material, such as the weak part of the material, the stress concentration, and the local deformation [77].

Rusch [78] first raised a universal model:(1)$$\sigma \left(\varepsilon \right)={\varepsilon E}_{f}\psi \left(\varepsilon \right).\;$$
where $E_{f}$ is the apparent Young’s modulus, and $\psi \left(\varepsilon \right)={A\varepsilon }^{m}+{B\varepsilon }^{n}$ is a dimensionless function of the compressive strain ε. Parameters A, B, m, and n are curve-fitting constant from a typical logarithmic plot of ψ(ε) against ε. This model was not accurate in describing the densification region.

Based on the uniaxial stress-strain test results, Liu and Subhash [79] proposed a six-parameter constitutive model of polyurethane foam to fit the linear elasticity, yield plateau, and densification region:(2)$$\sigma \left(\varepsilon \right)=A\left(\frac{e^{\alpha \varepsilon }-1}{B+e^{\beta \varepsilon }}\right)+e^{C}\left(e^{\gamma \varepsilon }-1\right)$$
where parameters A, B, α, and β are constant for a given initial density and strain rate. A is the scaling factor for yield stress; B plays a role of shifting the lower asymptote of the function. And parameters C and γ describe the densification region, varying C implies stretching or shrinking of the curve on the strain axis.

Avalle [80] improved the curve knee at the elastic and plateau region connection:(3)$$\sigma \left(\varepsilon \right)=A\left(1-e^{-\left(E/A\right)\varepsilon {\left(1-\varepsilon \right)}^{m}}\right)+B{\left(\frac{\varepsilon }{1-\varepsilon }\right)}^{n}$$
where A and E are the plateau stress and the elastic modulus, exponent m represents the connection between the elastic region and the plateau region. And parameters B and n describe the densification region.

Goga [77] proposed Maxwell model:(4)$$\sigma \left(\varepsilon \right)=e^{-\frac{k\varepsilon }{c}}\left(-1+e^{\frac{k\varepsilon }{c}}\right)c+\left[k_{p}+\gamma {\left(1-e^{\varepsilon }\right)}^{n}\right]\varepsilon$$
where k is equal to the elastic modulus; c is equal to the plateau stress; $k_{p}$ is the slope of the plateau region; and $k_{D}\left(\varepsilon \right)=\gamma {\left(1-e^{\varepsilon }\right)}^{n}$, which describes the densification region

Cousins [81] proposed a constitutive equation with foam strain rate effect:(5)$$\sigma =K{\left(1-\varepsilon \right)}^{-n\;\dot{\varepsilon }r}$$
where K is the plateau stress, n and r are constants used to simulate strain dependence and strain rate dependence respectively, typically in the range of 0 ≤ n ≤ 4 and 0 ≤ r ≤ 0.3. The constitutive equation can describe the mechanical behavior well in the strain range of 0.1–0.7, but it cannot fit the linear elasticity and densification region well.

Sherwood [82] considered the influence of the strain rate and proposed a stress-strain framework of cellular medium, as shown in the following formula:(6)$$\sigma \left(\varepsilon \right)=f\left(\varepsilon \right)M\left(\varepsilon ,\;\dot{\varepsilon }\right)$$
where $f\left(\varepsilon \right)=\overset{10}{\underset{i=1}{\sum }}A_{i}\varepsilon^{n}$ is the strain function,$\;A_{i}$’s are determined using a tenth-order polynomial fit. $M\left(\varepsilon ,\;\dot{\varepsilon }\right)={\left(\;\dot{\varepsilon }/{\;\dot{\varepsilon }}_{0}\right)}^{n\left(\varepsilon \right)}$ is the modulus function. ${\;\dot{\varepsilon }}_{0}$ is a quasi-static strain rate. n(ε)=a+bε is a linear function of strain, which is the slope of the relationship between the stress and strain rate [83]. a and b are material constants in the ranges of 0–0.0442 and 0.0343–0.145 [83].

Zhang [84] considered the influencing factors of temperature and strain rate to establish the uniaxial constitutive relationships of polypropylene foam, polyurethane foam and polystyrene foam according to the work of Nagy [83]. They extended the constitutive relationship to triaxial loading and successfully used it in a finite element program. The one-dimensional constitutive model is:(7)$$\sigma \left(\varepsilon \right)=\sigma_{0}\left(\varepsilon \right){\left(\frac{\;\dot{\varepsilon }}{{\;\dot{\varepsilon }}_{0}}\right)}^{n\left(\varepsilon \right)}\exp\left[-\frac{C_{1}\left(T-T_{r}\right)}{C_{2}+T-T_{r}}\right]$$
where $\sigma_{0}\left(\varepsilon \right)$ represents the nominal stress response at a constant quasi-static strain rate, $T_{r}$ is the glass transition temperature. The meaning of n(ε)=a+bε is reflected in Equation (6). $C_{1}$ and $C_{2}$ are Williams-Landel-Ferry (WLF) function coefficients to be determined from temperature effect experiments.

Jeong [85,86] added a modulus function (strain rate dependence) to the Avalle’s model [80], and proposed a constitutive model for the stress-strain relationship of cellular medium at low strain rates:(8)$$\sigma \left(\varepsilon \right)=\left\{A\left(1-e^{-\left(E/A\right)\varepsilon {\left(1-\varepsilon \right)}^{m}}\right)+B{\left(\frac{\varepsilon }{1-\varepsilon }\right)}^{n}\right\}M\left(\varepsilon ,\;\dot{\varepsilon }\right).\;$$
where the meanings of A, B, E, m and n are reflected in Equation (3) [80] and these parameters can be obtained at a quasi-static strain rate of 0.001 s^−1^. They thought the modulus function $M\left(\varepsilon ,\;\dot{\varepsilon }\right)$ worked as a scale factor of the shape function linearly varying along the strain axis between a reference strain rate (quasi-static strain rate 0.001 s^−1^) and another strain rate (0.1 s^−1^). And they proposed two modulus functions $M\left(\varepsilon ,\;\dot{\varepsilon }\right)$, as shown in Equations (9) [85] and (10) [86].
(9)$$M\left(\varepsilon ,\;\dot{\varepsilon }\right)=1+\left(a+b\varepsilon \right)\ln\left(\frac{\;\dot{\varepsilon }}{{\;\dot{\varepsilon }}_{0}}\right)$$
where a and b are obtained by linearly approximating of the experimental stress-strain data. ${\;\dot{\varepsilon }}_{0}$ is a quasi-static strain rate of 0.001 s^−1^. Another modulus function has different expressions of n(ε) depending on three strain regions. A cubic polynomial function of ε is used on the transition zone ($\varepsilon_{1}\leq \varepsilon \leq \varepsilon_{2}$), which consists of plateau region and densification region adjacent to the critical strain ($\varepsilon_{1}\leq \varepsilon_{\mathrm{cr}}\leq \varepsilon_{2}$). Linear functions of n(ε) are used for plateau region ($\varepsilon \leq \varepsilon_{1}$) and densification region ($\varepsilon \geq \varepsilon_{2}$). $c_{1}$, $c_{2}$, $c_{3}$ and $c_{4}$, can be obtained through boundary conditions at $\varepsilon =\varepsilon_{1}$ and $\varepsilon =\varepsilon_{2}$.
(10)$$M\left(\varepsilon ,\;\dot{\varepsilon }\right)=1+n\left(\varepsilon \right)\ln\left(\frac{\;\dot{\varepsilon }}{{\;\dot{\varepsilon }}_{0}}\right)$$
(11)$$n\left(\varepsilon \right)=\{\begin{array}{c} a_{1}+b_{1}\varepsilon \;\;\;\;\;\;\;\;\;\;\;\;\;\;\;\;\;\;\;\;\;\;\varepsilon \leq \varepsilon_{1}\;\\ c_{1}+c_{2}\varepsilon +c_{3}\varepsilon^{2}+c_{4}\varepsilon^{3}\;\;\;\;\varepsilon_{1}\leq \varepsilon \leq \varepsilon_{2}\;\\ a_{2}+b_{2}\varepsilon \;\;\;\;\;\;\;\;\;\;\;\;\;\;\;\;\;\;\;\;\;\varepsilon \geq \varepsilon_{2}\; \end{array}$$
(12)$$\left[\begin{array}{cccc} 1 & \varepsilon_{1} & \varepsilon_{1}^{2} & \varepsilon_{1}^{3}\\ 0 & 1 & 2\varepsilon_{1} & 3\varepsilon_{1}^{2}\\ 1 & \varepsilon_{2} & \varepsilon_{2}^{2} & \varepsilon_{2}^{3}\\ 0 & 1 & 2\varepsilon_{2} & 3\varepsilon_{2}^{2} \end{array}\right]\left[\begin{array}{c} c_{1}\\ c_{2}\\ c_{3}\\ c_{4} \end{array}\right]=\left[\begin{array}{c} a_{1}+b_{1}\varepsilon_{1}\\ b_{1}\\ a_{2}+b_{2}\varepsilon_{2}\\ b_{2} \end{array}\right]$$
where $a_{1}$, $a_{2}$, $b_{1}$ and $b_{2}$, are obtained by fitting the experimental stress-strain data at strain rate of 0.1 s^−1^. $c_{1}$, $c_{2}$, $c_{3}$ and $c_{4}$ are determined by solving the linear simultaneous Equation (12).

In the dynamic load, cellular media will inevitably encounter the impact of shock waves. Therefore, different shock wave models of cellular media are proposed to fit different types of stress-strain curves [46,87], as shown in Figure 4a. Reid and Peng [88] proposed a one-dimensional shock model. They assumed that the constitutive relationship of cellular media was independent on strain rate and obeyed the rigid-perfectly plastic-locking (R-PP-L) relationship. There were only yield stress and densification strain in the model. Lopatnikov [89,90,91] considered the effect of the elastic section of a cellular medium on the mechanical properties at low velocities on the basis of R-PP-L model, and proposed an elastic-perfectly plastic-rigid (E-PP-R) model, which took into account the elastic modulus. Zheng [92] proposed a rigid-linear hardening plastic-locking (R-LHP-L) model. A shock model under high-velocity impact and a transition model under medium-speed impact were established. In addition, Harrigan [93] used the rigid-softening-hardening (RSH) model and elastic-softening-hardening (ESH) model to fit the impact load on oak tree. Pattofatto [94,95] and Zheng [96] proposed a simple rigid-power-law harding (R-PLH) model to better evaluate the stress-strengthening level of impact enhancement. This model gave a closed or semi-closed solution of the physical quantities on the shock wave front.

Although the phenomenological constitutive models can fit the compressive stress-strain curves of cellular media well, they cannot explain the mechanism of the compression process.

### 3.2. Homogenization Algorithm Model

A cellular medium can be considered as a two-phase composite with matrix and pores. The homogenization algorithm model [97,98,99,100,101,102,103,104] combines a macro-scale and a micro-scale to predict the effect of microstructure on the overall properties of the material. The two scales are connected by a representative volume element (RVE). In a micro–macro-approach, at each macro-point $\bar{x}$, we know the macro-strain $\bar{\varepsilon }$ and need to compute the macro-stress $\bar{\sigma }$. At micro-level, the macro-point is viewed as the center of a RVE with domain ω and boundary ∂ω, as shown in Figure 4b. The Hill-Mandell condition, expressing the equality between energies at both scales, transforms the relation between macro-strains $\bar{\varepsilon }$ and macro-stresses $\bar{\sigma }$ into a relation between average strains 〈ε〉 and average stresses 〈σ〉 over the RVE, with $\bar{\varepsilon }=\langle \varepsilon \rangle$ and $\bar{\sigma }=\langle \sigma \rangle$. The matrix (phase 0) has volume $V_{0}$ and volume fraction $v_{0}=V_{0}/V$, where V is the volume of the RVE. The pores (phase 1) have a total volume $V_{1}$ and a volume fraction $v_{1}=V_{1}/V=1-v_{0}$ (subscript 0 refers to the matrix and 1 to the pores). The volume averages are defined as:(13)$$\langle f\rangle =\frac{1}{V}\int_{\omega }f\left(\bar{x},x\right)dV,\;\;{\langle f\rangle }_{\omega_{i}}=\frac{1}{V_{i}}\int_{\omega_{i}}f\left(\bar{x},x\right)dV_{i},\;\;i=0,1$$
where $\bar{x}$ is the macro-point, and x is the micro-point. The averages over ω (the entire RVE), $\omega_{0}$ (the matrix phase) and $\omega_{1}$ (the pores phase) are related by $\langle f\rangle =\left(1-v_{1}\right){\langle f\rangle }_{\omega 0}+v_{1}{\langle \varepsilon \rangle }_{\omega 1}$. The strain averages and stress averages can be written as [97,101]:(14)$${\langle \varepsilon \rangle }_{\omega }=\left(1-v_{1}\right){\langle \varepsilon \rangle }_{\omega_{0}}+v_{1}{\langle \varepsilon \rangle }_{\omega_{1}},{\langle \sigma \rangle }_{\omega }=\left(1-v_{1}\right){\langle \sigma \rangle }_{\omega_{0}}+v_{1}{\langle \sigma \rangle }_{\omega_{1}}$$

Under linear boundary conditions, the strain averages per phase are related by a strain concentration tensor $B^{\varepsilon }$ as follow:(15)$${\langle \varepsilon \rangle }_{\omega_{1}}=B^{\varepsilon }:\;{\langle \varepsilon \rangle }_{\omega_{0}}\;$$

To express this strain concentration tensor $B^{\varepsilon }$, assumptions should be made on the micro-mechanics. This model is based on the Eshelby tensor $S\left(I,C_{0}\right)$ and assumes that the strain field inside the inclusion (pore) is uniform and related to the macro-strain [103].
(16)$$B^{\varepsilon }={\left\{I+S:\left[{\left(C_{0}\right)}^{-1}:C_{1}-I\right]\right\}}^{-1}\;$$
where Eshelby tensor $S\left(I,C_{0}\right)$ depends on the geometry of the pore (I) and the stiffness tensor of the matrix $C_{0}$. The per-phase average strains are related to the macro-strain 〈ε〉 by:(17)$${\langle \varepsilon \rangle }_{\omega_{0}}={\left[v_{1}B^{\varepsilon }+\left(1-v_{1}\right)I\right]}^{-1}:\;\langle \varepsilon \rangle ,\;\;{\langle \varepsilon \rangle }_{\omega_{1}}=B^{\varepsilon }:\;{\left[v_{1}B^{\varepsilon }+\left(1-v_{1}\right)I\right]}^{-1}:\;\langle \varepsilon \rangle \;$$

We consider two-phase composites where the pores (phase 1) have the same shape, orientation and stiffness tensor $C_{1}$. Thus the macroscopic material behavior can be written in the form [30,97,101,102,103]
(18)$$\bar{\sigma }=\bar{C}:\bar{\varepsilon },\;\;with\;\;\;\;\;\;\bar{C}=\left[v_{1}C_{1}:B^{\varepsilon }+\left(1-v_{1}\right)C_{0}\right]:{\left[v_{1}B^{\varepsilon }+\left(1-v_{1}\right)I\right]}^{-1}\;$$
where $\bar{C}$ is the homogenized macro-stiffness tensor.

The homogenization model can predict heterogeneity by considering the volume composition of composite materials [97,100,101,102,103,104]. Compared with the finite element (FE) calculation which takes up much CPU computing time, the homogenization model can quickly calculate the stress, strain, and stiffness of composite materials. However, it cannot give the detailed strain and stress field in each phase, and the inclusions are limited to the ellipsoidal surface.

The homogenization model is only available when the matrix phase and the inclusion phase are both isothermal linear elastic materials. However, a cellular medium is not just a linear elastic material. To study the plasticity of material, it is necessary to adopt a progressive damage model. The progressive damage model of composite material belongs to the category of continuum damage mechanism. Different from the damage model that studies the evolution of a single macroscopic crack inside a composite material, the progressive damage model introduces damage variables to express the microdefects in the entire material [105,106,107,108]. These micro-defects can be considered to be continuously distributed inside the material. They will cause the actual internal stress of the material to be higher than the characteristic stress obtained by external measurement. It can be expressed that the stiffness of the material with defects is lower than the stiffness of the non-damaged material, as shown in Figure 4b. Based on the Kachanov-Rabotnov theory, the damage variable D is introduced to describe the constitutive relationship of the damaged material. The one-dimensional characterization stress σ and the real stress (macro-stress) $\bar{\sigma }$ can be simplified as:(19)$$\sigma =\left(1-D\right)\times \bar{\sigma }$$

Based on the equivalent strain, the constitutive relationship of the three-dimensional damaged material can be obtained. The three-dimensional characterization stress σ is:(20)$$\bar{\sigma }=\bar{C}:\varepsilon ,\;\sigma =C\left(D\right):\varepsilon$$
(21)$$\sigma =\left(1-D\right):\bar{\sigma }=\left(S^{-1}:\bar{S}\right):\bar{\sigma }$$
where $\bar{S}$ is the stiffness with no damage, and S is the characteristic damage stiffness related to the damage variable. D is a second-order tensor, and S is a fourth-order tensor. The stiffness matrix of the damaged material S(D) is
(22)S(D)=[S¯11111−D11S¯1122S¯1133000 S¯22221−D22S¯2233 0 0 0  S¯33331−D33 0 0 0   S¯12121−D12 0 0sym.   S¯23231−D23 0     S¯13131−D13]
where $D_{ij}$ is the element of the damage variable matrix D. ${\bar{S}}_{ijkl}$ is the element of the non- damaged material matrix $\bar{S}$.The progressive damage model describes the generation and evolution of damage variable D. Damage evolution is a function of plasticity: damage initiates at the onset of plasticity. The damage variable D in damage evolution is the key issue and foundation. It directly reflects the damage mechanism of the material. In damage mechanics, the damage variable D is used to describe the degradation process of material.
(23)D=φ(f)
where φ(f) is the damage evolution function, which reflects the law of the internal damage of the material. Generally, D=0 in the non-destructive state, and D=1 in the complete damage state.

The Hashin damage model describes damage factor D as a function of the Hashin failure criterion coefficient f. The model assumes that each phase of material obeys Weibull distribution. The damage evolution process is actually the process of strain energy release. In the process of strain energy release, the material will soften, and the macroscopic manifestation is the degradation of elastic modulus and the decrease of load-bearing capacity. The functional relationship between damage factor D and failure factor f can be defined as a variety of functional forms, so that the occurrence and evolution of damage can be described flexibly. The damage evolution function φ(f) includes: the step function damage evolution model Equation (24), power function damage evolution model Equation (25), linear softening damage evolution model Equation (26) and exponential function damage evolution model Equation (27) [97,102,103,104].
(24)$$\varphi \left(f\right)=\{\begin{array}{c} \;\;\;\;\;\;\;\;0\;\;\;\;\;\;\;\;\;\;\;\;\;\;\;\;\;\;\;\;\;\;\;\mathrm{if}\;f<f_{\min};\\ D_{\max}\;\;\;\;\;\;\;\;\;\;\;\;\;\;\;\;\;\;\mathrm{otherwise} \end{array}$$
(25)$$\varphi \left(f\right)=\{\begin{array}{c} 0\;\;\;\;\;\;\;\;\;\;\;\;\;\;\;\;\;\;\;\;\;\;\;\;\;\;\;\;\;\;\mathrm{if}\;f<f_{\min};\\ D_{\max}\times \frac{f^{\alpha }-f_{\min}^{\alpha }}{f_{\max}^{\alpha }-f_{\min}^{\alpha }}\;\;\;\;\;\;\;\;\;\;\;\;\;\;\;{\;\mathrm{if}\;f}_{\min}\leq f<f_{\max};\\ D_{\max}\;\;\;\;\;\;\;\;\;\;\;\;\;\;\;\;\;\;\;\;\;\;\;\;\;\;\;\;\;\;\;\;\;\mathrm{otherwise}.\;\;\;\;\;\;\;\;\;\; \end{array}$$
(26)$$\varphi \left(f\right)=\{\begin{array}{c} 0\;\;\;\;\;\;\;\;\;\;\;\;\;\;\;\;\;\;\;\;\;\;\;\;\;\;\;\;\;\;\mathrm{if}\;f<f_{\min};\\ D_{\max}\times \frac{f_{\max}}{f}\times \frac{f-f_{\min}}{f_{\max}-f_{\min}}\;\;\;\;\;{\;\mathrm{if}\;f}_{\min}\leq f<f_{\max};\;\\ D_{\max}\;\;\;\;\;\;\;\;\;\;\;\;\;\;\;\;\;\;\;\;\;\;\;\;\;\;\;\;\;\;\;\;\;\mathrm{otherwise}.\;\;\;\;\;\;\;\;\;\; \end{array}$$
(27)$$\varphi \left(f\right)=\{\begin{array}{c} \;\;0\;\;\;\;\;\;\;\;\;\;\;\;\;\;\;\;\;\;\;\;\;\;\;\;\;\;\;\;\;\mathrm{if}\;f<f_{\min};\\ D_{\max}\times \left(1-\exp\left(-\frac{f^{\alpha \beta }-f_{\min}^{\alpha \beta }}{e\beta }\right)\right)\;\;\;\;\mathrm{otherwise}. \end{array}$$
where f is the failure factor of the material, $f_{\min}$ is the minimum value, and $f_{\max}$ is the maximum value. $D_{\max}$ is the maximum value of material damage coefficient. α and β are the material response parameters.

Although the above models can describe the stress-strain relationship of cellular media well, they cannot explain the mechanism during stretching or compression. The constitutive models of high strain rate, cell size and doped particles cannot be obtained. Therefore, it is necessary to deeply investigate the constitutive models of loading conditions and cell structure of cellular medium.

## 4. Microstructure Mechanical Models of Cellular Medium

The deformation and evolution of a cellular medium’s internal structure is complex, and there are many problems in traditional experimental methods. Numerical simulation based on fine analysis model has become a valid research method.

### 4.1. Single Cell Model

The single cell model is based on analyzing the deformation mechanism of a tiny element (a cell) under load, as shown in Figure 5. Gibson and Ashby [28] first proposed a GA model suitable for open-cell and closed-cell foam, which simplified the cellular medium into countless cells composed of cell edges and cell walls. Under pressure, the cell edges bend, and the cell walls stretch. The relationship between the relative modulus $E_{r}$, and relative strength $\sigma_{r}$, and relative density $\rho_{r}$ can be obtained by:(28)$$E_{r}=\phi^{2}\rho_{r}^{2}+\left(1-\phi \right)\rho_{r}$$
(29)$$\sigma_{r}=\phi^{1.5}\rho_{r}^{1.5}+\left(1-\phi \right)\rho_{r}$$
where ϕ is the volume fraction of cell edge, which is a correction term of 0–1. Since the value of ϕ is unrelated to density only in an ideal state, it is difficult to determine the value for actual closed-cell materials, especially when the density is high. Therefore, there are errors in the Gibson-Ashby model.

Choonghee [109,112] investigated the mechanical strength of open-cell PMMA foam. They assumed that the open-cell foam was composed of many equal cubic frames, and the simplified structural unit is shown in Figure 5c. Choonghee obtained the relationship between relative elastic modulus and relative density:(30)$$E_{r}=C_{0}{\left(0.1459+0.7082\rho_{r}\right)}^{4}$$
where $C_{0}$ is a constant fitted by experimental data. Chen [110] explored the constitutive model of closed-cell foam. The basic unit is similar to the open-cell structure unit, as shown in Figure 5d. The cell wall is added to the open-cell model. The thickness of cell wall $t_{f}$ is equivalent to the thickness of cell edge $t_{e}$ both is t/2, and the length of cell edge is also *l*. 

According to the Gibson-Ashby model, $E_{r}$ of closed-cell constitutive model can be calculated as:(31)$$E_{r}=\left(\frac{3A_{r}{}^{2}-2A_{r}{}^{3}}{1-{\left(1-A_{r}\right)}^{3}}\right)A_{r}{}^{4}+\beta \left(1-\frac{3A_{r}{}^{2}-2A_{r}{}^{3}}{1-{\left(1-A_{r}\right)}^{3}}\right)\frac{A_{r}}{2}$$

$A_{r}$ can be calculated as:
(32)$$A_{r}=1-{\left(1-\rho_{r}\right)}^{\frac{1}{3}}$$

When the relative density is high, the closed-cell constitutive model prediction is slightly lower than the measured result. They pointed out that when the thickness of cell wall is different from the thickness of cell edge, the phenomenon of “too low” prediction will appear, and the cell structure tends to be round rather than square. Chen believed that when the relative density was lower than 0.3, the relative strength of cellular medium could be predicted well. Triawan [111] proposed an improved GA model and derived elastic modulus empirical equations under different uniaxial loadings according to experimental results.

On this basis, researchers developed tetradecahedron models (Kelvin model) [113,114,115] and octahedron models [116,117,118], as shown in Figure 6. These single cell models provide an easy method to analyze the mechanical properties, but the simple shape and regular permutation of them are too idealistic and far from the realistic structure. They cannot represent the mechanical properties of the entire cellular medium.

### 4.2. Multi-Cell Model

Many scholars have conducted research on the finite element (FE) method. For cellular media, modeling is important. Only when a model more closely describes real conditions, can the calculation agree with experimental values.

Some researchers [119,120,121,122,123,124] established 2D multi-cell models to investigate the mechanical properties of cellular media, as shown in Figure 7. Sun [120] thought that entrapped gas hardly affects plateau stress, but it did affect the deformation modes. In the densification region, the entrapped gas significantly enhanced crushing stress, as shown in Figure 8a,b. Generally, the effect of entrapped gas on the hexagonal cells is more important than that on the circular cells, indicating that it is dependent on the cell morphology.

Zhang [119] established a circle arc (c-a) model, as shown in Figure 7c. Under quasi-static conditions, the cellular medium deformed into an X-shape, and the broken zone spread until it was completely densified. At moderate impact velocities, V-shaped deformation can be found. At high velocities, the crushing zone is positioned high on the impact side, which is an I-shaped deformation, and then spreads along the crushing direction, as shown in Figure 8c.

Mukhopadhyay [122,123,124] investigated the mechanical properties (Young’s modulus, Poisson’s ratio, and shear modulus) of 2D honeycomb material based on a regular hexagon-cell model and irregular hexagon-cell model. They considered the viscoelasticity and the irregularity of the model, and found that the Young’s modulus and shear modulus depend on the viscoelastic parameters, while the Poisson’s ratio was independent on the viscoelastic parameters and frequency. They put forward the expressions of elastic modulus under static loading and steady-state vibration conditions.

Some researchers established 3D multi-cell models to investigate mechanical properties of cellular medium, as shown in Figure 9. Song [125] used the finite element software ABAQUS to simulate the dynamic mechanical behavior of closed-cell foam with the Voronoi models [61,126,127,128,129,130] and tetradecahedron models [131,132,133]. At low impact velocities, during the initial compression process, plastic deformation first begins to appear in the middle of the tetradecahedron model. At high impact velocities, cell crushing propagates downwards from the impacted surface, which exhibits an “I” shaped deformation mode, as shown in Figure 10a.

Li [135] and Xue [33] concluded that cellular media showed different deformation modes under different impact velocities. They observed that at low impact velocities, the deformation zones were randomly distributed, while at medium velocities there were transitional deformation modes. When the impact velocity was high enough, the broken zones were locally concentrated in the impacted area and then spread along the direction of propagation. Figure 10e compares the strain rate enhancement and inertia enhancement at different impact velocities. In the random mode, the stain rate enhancement decides the strength enhancement. In the band front mode, the inertia enhancement decides the strength enhancement.

Dabo [134] established an ad-hoc model to generate closed-cell foam microstructures, as shown in Figure 9f. Figure 10c showed that the ad-hoc model was the closest to the real microstructure and compressive behavior of the cellular medium material. The ad-hoc model adopted a dynamic method that considered the effect of temperature on the material characteristics. It can simultaneously consider cell nucleation, growth through gas expansion, and the interactions between growing cells.

De Giorgi [113,136] developed a randomly distributed ellipsoidal cell model based on the Kelvin cell model, as shown in Figure 9c. Fang [137,138] presented a three-dimensional (3D) mesoscopic model of closed-cell foam material [139].

These models assume equal thickness of cell wall, which is inconsistent with reality. And they are affected by random factors, and the algorithm stability is not high. There is unsatisfactory practical application of these models, and they can only provide a simple analytical method of cellular media.

To truly describe the microstructure of cellular medium, some researchers use computed tomography (CT) or X-ray scanning technology to characterize the cellular medium, as shown in Figure 11. This technology can display the material microstructure layer by layer intuitively and nondestructively. The 3D microstructure of cellular medium can be further acquired by image reconstruction technology, which restores the topological structure to the greatest extent [60,130,131,132,133,134,135,136,137,138,139,140,141,142,143,144,145,146,147,148,149]. It has authenticity unmatched by other methods.

Huang [141] established the finite element model of cenosphere epoxy syntactic foams in ABAQUS/Explicit software using X-ray scanning technology. As shown in Figure 12a, the stress is concentrated near the top and bottom of any single cenosphere as well as the junction between adjacent cenospheres. Microcracks tend to initiate in the cenosphere connection area of the matrix. Then, the microcracks spread and merge with other microcracks and voids left by the broken cenospheres to form macrocracks.

Wang [150] used micro-CT technology to numerically model the yield characteristics of aluminum (Al) foam [151,152,153]. Figure 12b presents deformation modes under compression and tension. During the compressive process, the plastic hinges are formed first, and then the plastic collapse arises in the weak area [143]. During the tension process, crack first occurs at the cell walls in the weak area, then spreads the load, and then completely fractured.

Kader [34] developed a FE model based on micro-CT to study the elastoplastic behavior of shock wave propagation and the cell collapse mechanisms of aluminum foams. As shown in Figure 12c, a sudden rise of free surface speed was discovered at 845 m/s, indicating the development of the shock wave. It describes elastic and plastic wave propagation characteristics of the impact load in the loading direction. The elastic wave front tagged with a red line is migrating in the direction of shock. The result shows that the cell collapses in an uneven manner and continue to collapse as the waves move.

However, the modeling algorithm based on X-ray and CT is complicated, with many intermediate processing steps, and often need large-scale commercial software. The cost of using X-ray or CT scanning technology is high, which limits its wide application.

## 5. Discussion

At present, researchers have performed much work characterizing mechanical properties, establishing the constitutive relationship and microstructure model, and optimizing current cellular media designs. This paper reviewed the mechanical properties characterization of cellular media, and provided the latest comments of key mechanical models, including classifying these models based on modeling scale and methodology, tracking the historical progress of these models, and discussing their scope of application, advantages and disadvantages.

The effects of cell size, matrix, relative density, strain rate and testing methods on the mechanical properties of cellular media were systematically investigated through quasi-static mechanical testing and dynamic mechanical testing. The investigation on mechanical properties of cellular media under explosive loading and ultra-high strain rates involves complex stress waves, and systematic analysis methods and influence laws have not yet been obtained.The constitutive model can quickly and accurately describe the stress-strain relationship of a cellular medium, and obtain mechanical parameters (elastic modulus, plateau stress, densification strain, etc.). However, it cannot explain the microstructure deformation mechanism during stretching or compressing. The constitutive model with cell parameters (cell size and cell shape), high strain rate (greater than 10^4^ s^−1^) and explosive loading needs further study.The microstructure model can not only obtain the stress-strain curve, but also explain the microstructure deformation mechanism of cellular media, such as cell deformation, weak parts of the material, stress concentration and local deformation. However, most of the models are based on various algorithms and describe the local microstructure (RVE) of cellular media, which is greatly affected by random factors and is not stable. It is a great challenge to establish a stable, low-cost, real and effective microstructure model of cellular media.Molecular dynamics, phase field method and peridynamics can be used to establish nanoscopic models to investigate the structural evolution and properties of molecules and atoms in cellular media. Due to these methods involve nanoscale, simulation results and experimental characterization results are not comparable, and the scale range is small and the calculation amount is large, so they are not introduced in this paper.

The research of cellular media is growing rapidly and has been widely used in fields of packaging boxes, transportation, daily necessities, industrial manufacturing, and aerospace. There will be more applications in the future:The deformation mode and failure mechanism of cellular media under explosive loading and ultrahigh strain rates, gradient cellular material (gradient mode) structural design and performance optimization [18,57,71,154,155,156,157,158,159], and functional foam composite materials (e.g., doped metal particles, carbon nanotubes, and graphene) [21,160,161,162,163] are the keys to future research on cellular media.Expand the mechanical constitutive models of cellular media to suitable for various complex loads, and propose more constitutive models to understand the multifunctional performance of cellular media.Novel design of cellular metamaterials with unprecedented properties and advanced manufacturing technology [164,165,166,167].Molecular dynamics [168,169,170], phase field method [171,172,173], peridynamics [174] combined with multi-physics coupling calculations can obtain the nanoscale structural evolution and multifunctional performance of cellular media.Multifunctional applications for potential industrial applications, such as: high-performance wave absorption characteristics [175,176,177], electromagnetic interference shielding [162,178,179,180], adsorption thermodynamics and dynamics characteristics [181], and can be used in medical care, satellite communications, electronic equipment, national defense security and other fields in the future [180,182,183,184,185]. And conduct more experiments to verify potential industrial applications.

## 6. Conclusions

This paper mainly focuses on the research progress of cellular medium materials in recent years, introduces the mechanical properties characterization of cellular media, and provides mechanical models to describe their stress-strain relationship. The phenomenological constitutive models and the homogenization algorithm models can quickly obtain the stress-strain relationship of cellular media, but they cannot investigate the mechanism of the microstructure deformation. The microstructure models can not only obtain the stress-strain curve, but can also observe the microstructure deformation of a cellular medium. However, these models are not stable and require a large amount of calculation. Molecular dynamics, the phase field method, and peridynamics are the research focus of cellular media at the molecular scale and atomic scale. This paper provides a reference and a compilation of important documents for researchers pursuing the same research direction, and points out the focus of the research and potential application directions.

## Figures and Tables

**Figure 1 polymers-13-03283-f001:**
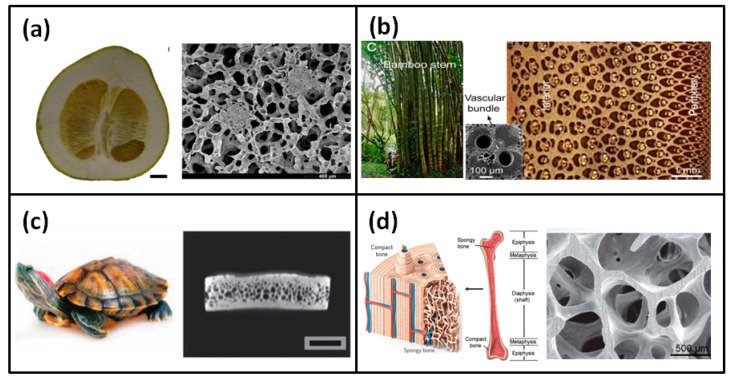
Cellular medium materials in nature (**a**) grapefruit peel, (**b**) bamboo, (**c**) turtle shell, and (**d**) bone [2,18].

**Figure 2 polymers-13-03283-f002:**
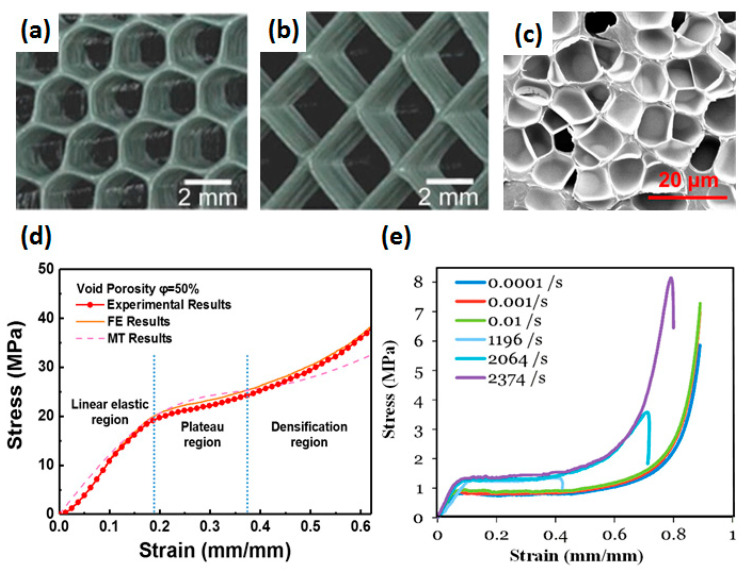
Artificial cellular medium (**a**) square honeycomb structure, (**b**) hexagonal honeycomb [2,29], (**c**) foaming materials [30], and stress-strain curves of cellular medium under (**d**) quasi-static compression [30] and (**e**) dynamic compression [31].

**Figure 3 polymers-13-03283-f003:**
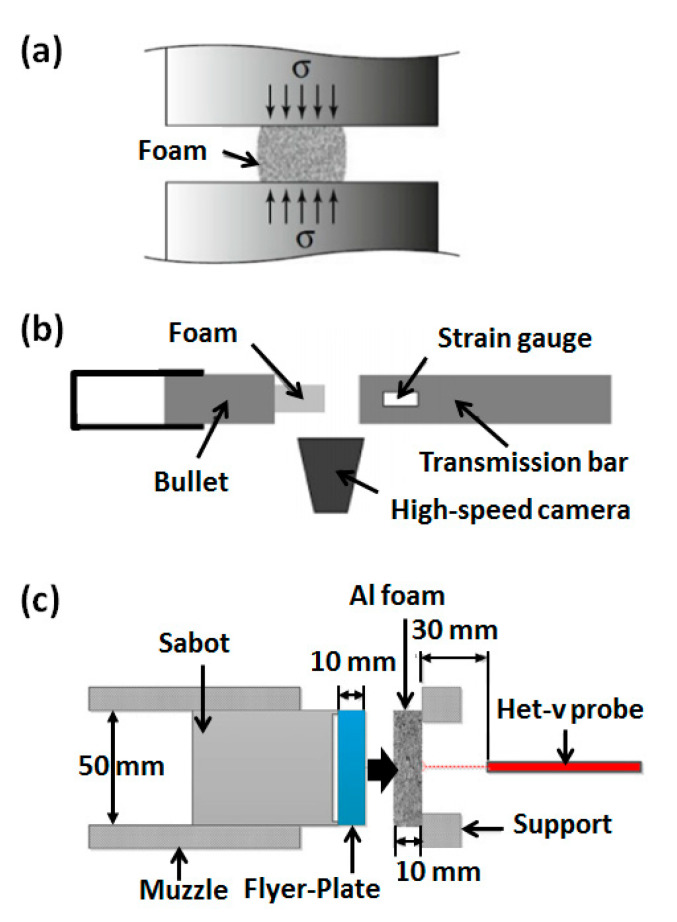
Schematic diagram of (**a**) quasi-static loading, (~10^0^ s^−1^) [32], (**b**) dynamic impact test (10^1^–10^5^ s^−1^) [33], and (**c**) flyer-plate impact test (~10^6^ s^−1^) [34].

**Figure 4 polymers-13-03283-f004:**
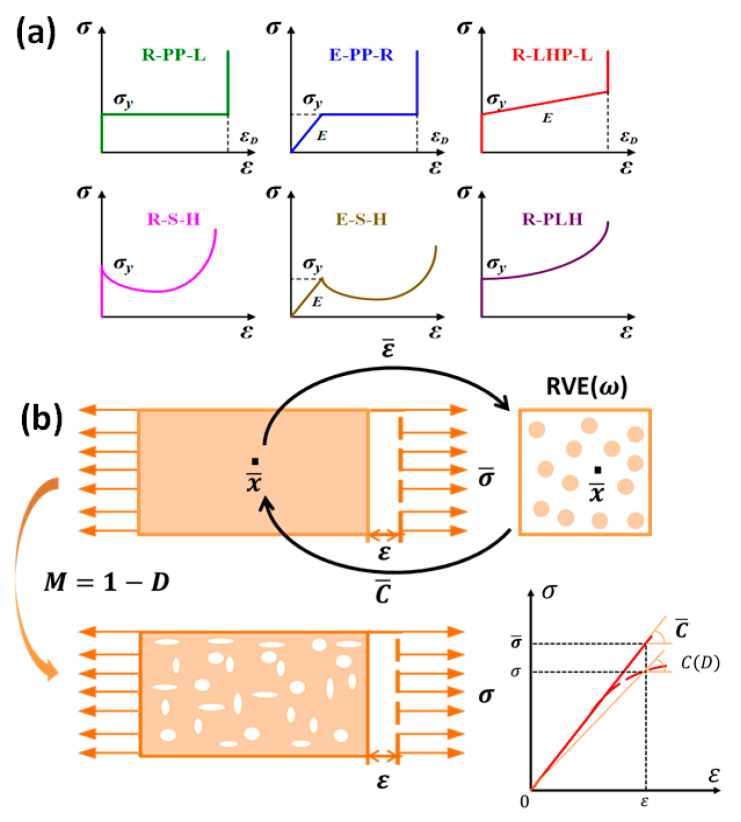
(**a**) Simplified one-dimensional shock models for cellular medium [46,87]; (**b**) basic model of homogenization algorithm model and progressive damage model [30,97].

**Figure 5 polymers-13-03283-f005:**
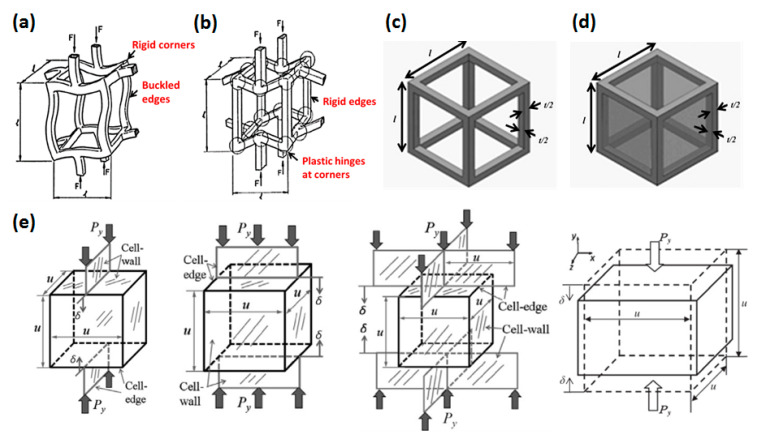
Deformation mechanism (**a**) elastic collapse and (**b**) plastic collapse [28], and single cell model (**c**) open-cell unit [109], (**d**) closed-cell unit [110], and (**e**) cell model under different uniaxial loadings [111].

**Figure 6 polymers-13-03283-f006:**
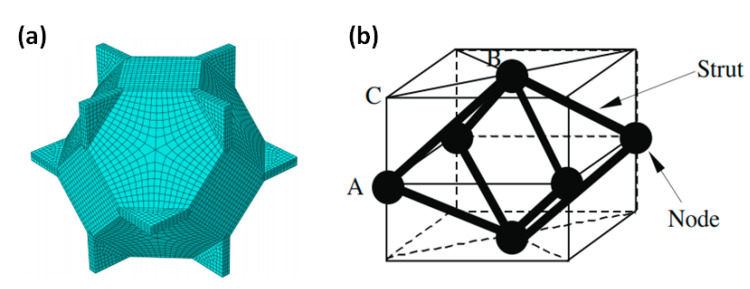
Improved single cell model (**a**) tetradecahedron model (Kelvin model) [115] and (**b**) octahedron model [116].

**Figure 7 polymers-13-03283-f007:**
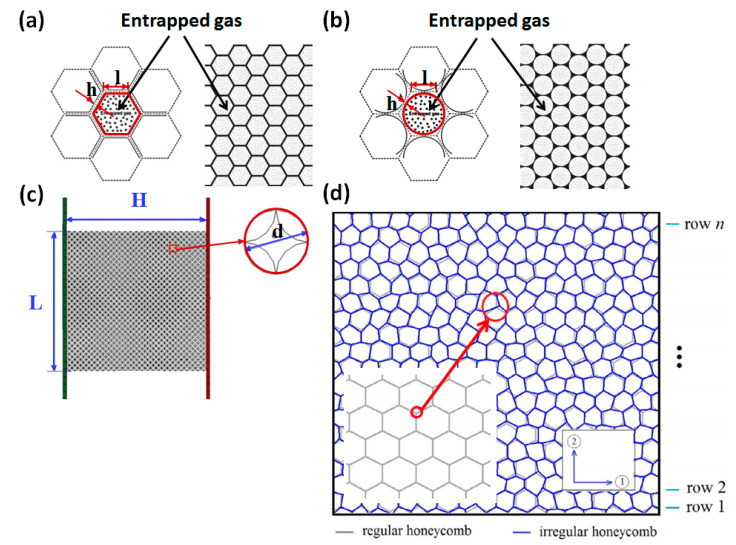
2D multi-cell models: (**a**) hexagonal-cell model [120]; (**b**) circular-cell model [120]; (**c**) circle arc (c-a) model [119]; (**d**) irregular hexagonal-cell model [122].

**Figure 8 polymers-13-03283-f008:**
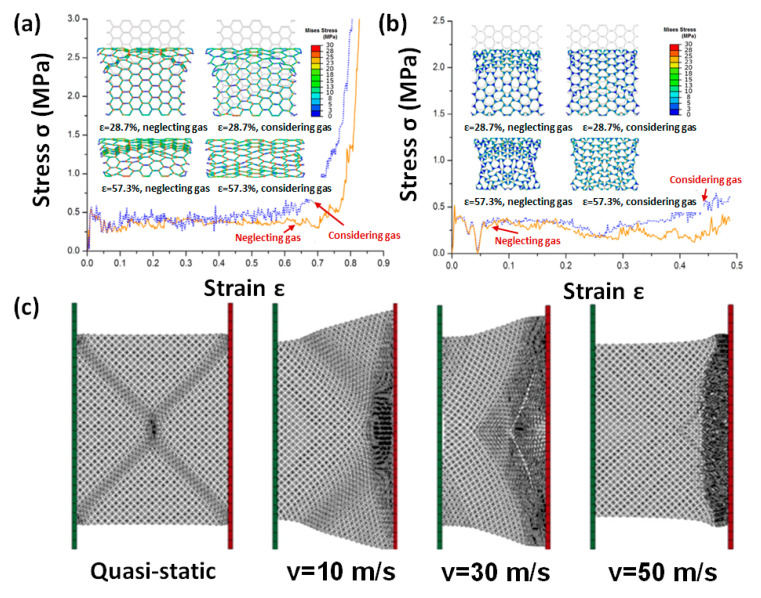
(**a**) Stress-strain curves of hexagonal-cell model at 100 s^−1^ [120], (**b**) stress-strain curves of circular-cell model at 100 s^−^^1^ [120], and (**c**) deformation mode of c-a model at different impact speeds [119].

**Figure 9 polymers-13-03283-f009:**
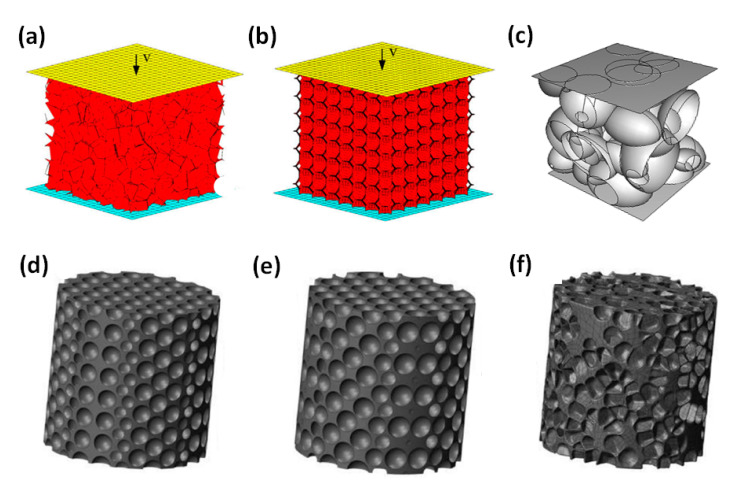
3D multi-cell models: (**a**) Voronoi model [33]; (**b**) tetradecahedron model [33]; (**c**) ellipsoidal-cell model [113]; (**d**) body-centered cubic (BCC) model [134]; (**e**) face-centered cubic (FCC) model [134]; (**f**) “Ad-hoc” model [134].

**Figure 10 polymers-13-03283-f010:**
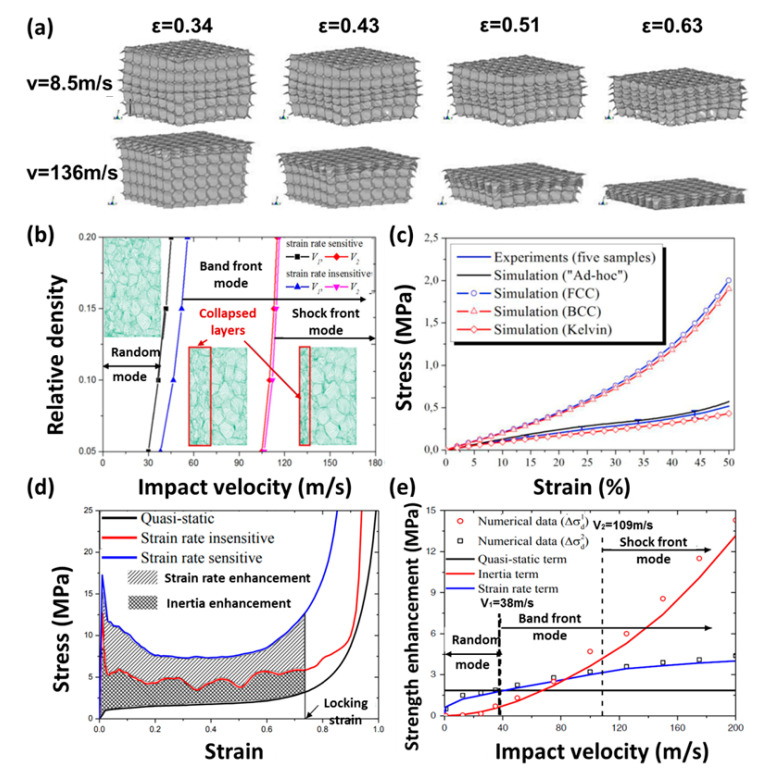
(**a**) Deformation process of tetradecahedron model at 8.5 m/s and 136 m/s [125], (**b**) deformation process of Voronoi model at different impact velocities [33], (**c**) stress-strain curves of Ad-hoc model, BCC model, FCC model and Kelvin model [134], (**d**) stress-strain curves of metal foam at 90 m/s [33], and (**e**) crushing strength under various impact velocities [33].

**Figure 11 polymers-13-03283-f011:**
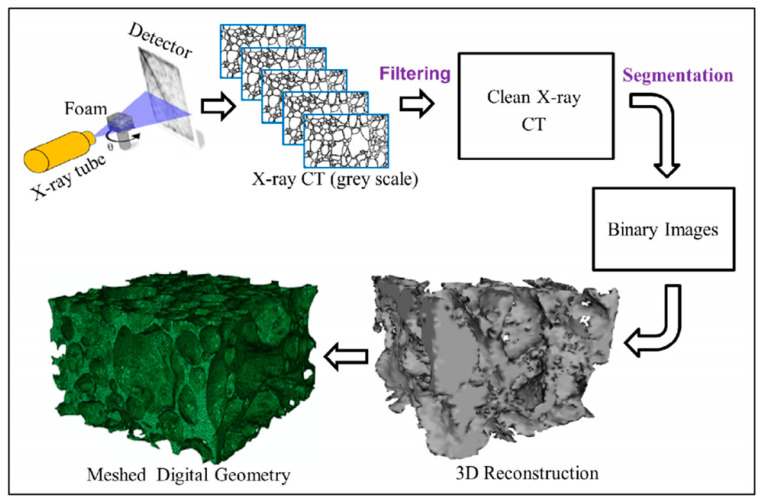
Flow chart of foam geometry development based on micro-CT: a flat panel detector and micro-focus X-ray source of 100 kV were used for imaging. The foam specimen was placed on a motor controlled rotating stage and radioscopic projections were taken after each degree of rotation. And the grey projections were transferred to binary images and processed to reconstruct full 3D digital foam geometry. Tetrahedral elements were used to mesh the geometry due to complex geometrical structure [34].

**Figure 12 polymers-13-03283-f012:**
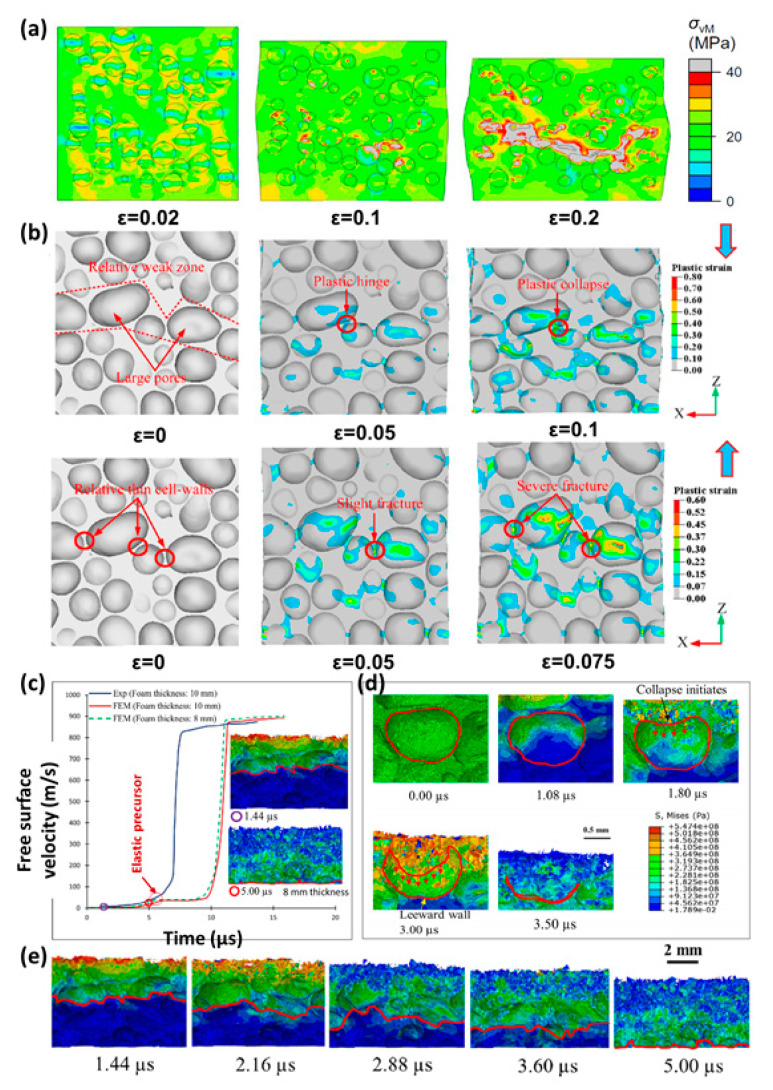
(**a**) Predicted von Mises stress distribution and failure process during uniaxial compression [141], (**b**) deformation patterns under uniaxial compressive loading and tension loading [150], (**c**) comparison of free surface velocity–time curve between experiment and FE simulations for 845 m/s flyer plate velocity, and the visual wave propagation is also correlated with velocity–time curve [34], (**d**) cell collapse mechanism with time for 845 m/s flyer plate velocity, and the red line indicates the periphery of a single pore [34], and (**e**) elastic wave propagation characteristic through a 8 mm thickness foam at 845 m/s [34].

**Table 1 polymers-13-03283-t001:** Performance comparison of various cellular medium mechanical models.

Models	Advantages	Disadvantages
Phenomenological constitutive model	Can obtain the mathematical expression between stress and strain by fitting experimental data; Can study the effect of void porosity and strain rate on the stress-strain curve of foam material; Can obtain important parameters in the compressive stress-strain curve of foam: linear elasticity (elastic modulus), yield plateau (yield stress), and densification region (densification strain).	♦ Cannot consider the effect of the microstructure (cell size, cell shape and cell distribution) on the stress-strain curve of the foam material; ♦ Cannot describe the stress-strain relationship at higher strain rates (10^5^ s^−1^~) or under explosive loading; ♦ The densification region fits poorly at high relative density (low void porosity).
Homogenization algorithm model	Considered as a multiphase composite of matrix and pores; Can quickly calculate the stress, strain, and stiffness.	♦ Cannot give the detailed strain and stress field in each phase (matrix or pores); ♦ Cells are limited to the ellipsoidal topological structure.
Single cell model	Use the deformation mechanism of a single cell under load to represent the deformation mechanism of the entire material; Can obtain relative modulus and relative strength of the foam.	♦ Fit poorly when the relative density is higher than 0.3; ♦ The cell shape is too simple and far from the realistic structure.
Multi-cell model	Can observe deformation process of the microstructure, local stress and strain; Modeling with algorithms.	♦ Complicated modeling algorithm; ♦ Need large-scale commercial software; ♦ Affected by random factors, and the algorithm stability is not high; ♦ High cost.

**Table 2 polymers-13-03283-t002:** Performance comparison of matrix material, relative density, cell size and strain rate.

Cellular Medium	Relative Density	Cell Size	Strain Rate	Modulus	Strength	Toughness	Performance	Ref.
Brittle carbon foam	0.029–0.04	0.49–4.9 mm	-	45–62 MPa	-	0.04–0.05 MPa$\sqrt{m}$	Mechanical properties are not significantly affected by cell size.	[35]
Polycarbonate (PC) foam	0.56	2.8–37.1 μm	-	836–978 MPa	25.6–29.8 MPa	7.02–10.22 J/cm^3^		[36]
Polyetherimide (PEI) foam	0.75–0.9	2–4 μm 30–120 nm	-	1929–2628 MPa	64.1–87.7 MPa	17.6–103.1 J/cm^3^		[37]
Polyurethane (PU) foam	0.98	0–800 μm	-	-	0.06–0.11 MPa	-	Compressive strength increased as the cell size (200–800 μm) increased, or as the cell size (0–200 μm) decreased.	[38]
3D hierarchical foam	-	0–55 nm	-	0.5–1 GPa	0.8–13 MPa	-	Mechanical properties increased as the cell size decreased.	[39]
Polymethyl methacrylate (PMMA) foam	0.5	0.2–11μm	-	577–791 MPa	-	0.13–0.44 J/cm^3^		[40]
Aluminum (Al) foam	0.02–0.2	3–25 mm	10^−4^–10^3^ s^−1^	1–700 MPa	1.5 MPa	-	Mechanical properties depend on relative density. Mechanical properties hardly changed by strain rate.	[41,42]
Expanded polystyrene (EPS) foam	0.04–0.11	130–320 μm	-	6.5–32.3 MPa	0.19–0.7 MPa	-		[43]
Al foam	0.15–0.29	0.8 mm	10^−3^–2600 s^−1^	-	5–22 MPa	-	Mechanical properties depend on relative density and are strain rate sensitive. Higher relative densities are more sensitive to strain rate. Strain rate hardening phenomenon.	[44,45,46]
Expanded polypropylene (EPP) foam	0.03–0.17	0–0.4 mm	10^−2^–1500 s^−1^	0.2–4 MPa	0.2–2.7 MPa	-		[47]
Polyvinyl chloride (PVC) foam	0.04–0.18	300–450 μm	10^−4^–2000 s^−1^	74–400 MPa	1.08–8.65 MPa	-		[31]
EPS foam	0.021–0.044	-	2.68–280 s^−1^	2.7–4.8 MPa	0.171–0.478 MPa	-		[48]

## Data Availability

Data available in a publicly accessible repository.

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
