# Peer review of "A Review on Mechanical Models for Cellular Media: Investigation on Material Characterization and Numerical Simulation"

_polymers, 2021, doi:10.3390/polym13193283_

Round 1

Reviewer 1 Report

The authors reviewed some progress on the development of mechanical models for cellular media. They introduced macroscopic constitutive laws and homogenized models as well as single or multiple cell model describing detailed microscopic structures. They raised some examples and discussed the validity and problems to be resolved. Finally, they proposed several topics for future study.

The manuscript is well-written and previous results can be easily followed. Only a minor error can be found as below. If the authors correct it, the reviewer will accept the manuscript for publication in Polymers.

In Table 1, “105 s-1” is correct? Isn’t it 105 s-1 ?

Author Response

Comments and Suggestions for Authors

The authors reviewed some progress on the development of mechanical models for cellular media. They introduced macroscopic constitutive laws and homogenized models as well as single or multiple cell model describing detailed microscopic structures. They raised some examples and discussed the validity and problems to be resolved. Finally, they proposed several topics for future study.

The manuscript is well-written and previous results can be easily followed. Only a minor error can be found as below. If the authors correct it, the reviewer will accept the manuscript for publication in Polymers.

In Table 1, “105 s-1” is correct? Isn’t it 105 s-1 ?

Response. Thank you for your suggestion. In Table 1, “105 s-1” is “105 s-1” and I have already modified in the manuscript.

Reviewer 2 Report

Title: "A review on mechanical models for cellular medium: investigation on material characterization and numerical simulation   "

This manuscript  reviews current key mechanical models for cellular medium, trying to track their evolution from inception  to the latest version. In particular, these models are categorized in terms of their mechanical modeling methods.  The authros claim that this work focuses on the importance of constitutive relationships and microstructure models in stud- 
ying mechanical properties and optimizing structural design. In addition, the author claim that the key issues and future directions 
are also discussed.

General comment. Unfortunately, this work should be deeply reworked, some mistakes should be corrected and its value should be better underlined in a  novel version of the manuscript which perhaps will be able to meet the quality requirements of this journal.

Some detailed comments:

Table 1. Performance comparison of various cellular medium mechanical models.

*) This table is not totally clear, then it should be reworked and improved.
In addition, the description of "features" and "Weaknesses" should be more precise.

Figure 3. Equipment and schematic diagram of (a) universal testing machine (quasi-static, ~100 s-1) 123
[32], (b) Split Hopkinson Pressure Bar (SHPB, 101-105 s-1) [33], and (c) light gas gun (106 s-1~) [34]

*) The authors should improve this figure and provide high resolution figures.
Please improve also the caption.

Table 2. Performance comparison of various cellular medium mechanical models.

*) This table with relevant information should be better presented. Please rework accordingly

3.1. Phenomenological constitutive model 216
The phenomenological constitutive model is the expression between stress and 217
strain obtained by fitting experimental data. It can be used to describe physical phe- 218
nomena, but it cannot explain the mechanism [77]. 21

*) The authors should explain what is the meaning of "is the expression between stress and 
strain" as well as the meaning of "it cannot explain the mechanism".
Perhaps they should express also which is the mechanism involved. 
In materials science there are several descriptions of the behaviour of materials and the  phenomenological equations are one of them. Nevertheless, the crucial points of this methods are totally neglected in these lines. Please rework.

*) Each formula presented should also describe in detail the meaning of all quantities for interested readers. Please rework all this section accordingly. Please explain also the physical meaning of all parameters.

 3.2. Homogenization algorithm model

 *) This paragraph is not clearly presented (in particular Eqs. 3.12-3.20 need to be clearly explalned in details). Please rework accordingly.

 Figure 6. Single cell model (a) tetradecahedron model (Kelvin model) [113,114] and (d) octahedron 364
model [115]

*) This figure should be improved as well as its caption

Figure 7. 2D multi-cell models: (a) hexagonal cells and circular cells [119]; (b) circle arc (c-a) model [118]; (c) bubble 376
growth 2D model [120]

Figure 8. (a) Dynamic crushing of hexagonal cells and circular cells at 100 s-1, (b) dependences of 385
stress enhancement and gas pressure on nominal strain [119], and (c) deformation of c-a model at 386
different impact speeds [118]

Figure 9. 3D multi-cell models: (a) Voroni model and tetradecahedron model [33,121], (b) “Ad-hoc” models with different 395
porosities and polydispersities [130], (c) bubble growth 3D model [120], (d) Body-centred cubic (BCC), Face-centred cubic 396
(FCC), Kelvin's tetrakaidecahedron structure [130], and (e) random model based on ellipsoidal cells [112]

Figure 10. Deformation modes of (a) Voroni model and tetradecahedron model at 8.5 m/s [121], (b) comparison of Voroni 407
model and sample at 30 m/s and 70 m/s, (c) Voroni model at different impact velocities and crushing strength under 408
various impact velocities [33], (d) deformation of ad-hoc model, and (e) compressive behavior of foams (Ad-hoc, BCC, 409
FCC, Kelvin, and samples) [130]

Figure 11. (a) Flow chart of foam geometry development based on micro-CT, (b) digital foam geometry and sectional 431
view [34], (c) the process to obtain the foam model and cell pore volume distribution and cell size volume ratio [146], and 432
(d) X-ray microtomographic image and geometrical model [137] 43

Figure 12. (a) Failure process during uniaxial compression [137], (b) deformation patterns and 443
stress-strain curves under uniaxial compressive loading and tension loading [146], and (c) cell col- 444
lapse mechanism , free surface velocity-time curves, and elastic wave propagation characteristic 445
through a 8 mm thickness foam at 845 m/s [34]

*) All these figures should be provided in high resolution and all the captions should be reworked to explain in details all the provided  images. When panels are used (as in this case) a detailed caption should be provided to the interested readers. Please, check also the correctness of names and citations. What is the meaning of "Voroni" ?????

lines:"5. Conclusions 465
At present, researchers have performed much work on the characterizing mechani- 466
cal properties, establishing the constitutive relationship and microstructure model, and 467
optimizing cellular medium designs. This paper introduced the mechanical models of 468
cellular medium, classified them according to their modeling methods, tracked the his- 469
torical progress of key models, and summarized main achievements in this field. We 470
focused on the importance of constitutive relationships and microstructure models in 471
studying mechanical properties and optimizing structural design. Although there has 472
been decades of research on the mechanical properties of cellular medium, there is still a 473
large number of key issues to be resolved. 474
• Although the constitutive model can depict the stress strain relationship of cellular 475
medium well, it cannot explain the microstructure deformation mechanism during 476
stretching or compressing. 477
• The phenomenological constitutive model with cell parameters (cell size and shape) 478
and a high strain rate (greater than 100 s-1) needs to be further investigated. 479
• The single cell model and multi-cell model can explain the microstructure defor- 480
mation mechanism of cellular medium. However, most of the models are based on 481
various algorithms and describe the local microstructure (RVE) of cellular medium, 482
which is greatly affected by random factors, and the stability is not high. 483
• Researchers should investigate how to build a true and effective microstructure 484
model of cellular medium and ensure the stability of the algorithm while reducing 485
its cost. 486
• The deformation mode and failure mechanism of cellular medium under explosive 487
loading and ultrahigh strain rates, gradient cellular material (gradient mode) struc- 488
tural design and performance optimization [18,57,71,150-155], and functional foam 489
composite materials (e.g., doped metal particles, carbon nanotubes, and graphene) 490
[21,156-159] are the keys to future research on cellular medium. 491
6. Future directions 492
The research of cellular medium is growing rapidly and has been widely used in the 493
fields of packaging boxes, transportation, daily necessities, industrial manufacturing, and 494
aerospace. There will be more application in the future: 495
• Expand the mechanical constitutive models of cellular medium to suitable for vari- 496
ous complex loads, and propose more constitutive models to understand the mul- 497
tifunctional performance of cellular medium. 498
• Novel design of cellular metamaterials with unprecedented properties and ad- 499
vanced manufacturing technology [160-164]. 500
• Multifunctional applications for potential industrial applications, such as: 501
high-performance wave absorption characteristics [165-167], electromagnetic inter- 502
ference shielding [158,168-170], adsorption thermodynamics and dynamics charac- 503
teristics [171], and can be used in medical care, satellite communications, electronic 504
equipment, national defense security and other fields in the future [170,172-175]. 505
And conduct more experiments to verify potential industrial applications. 506"

*) This work completely lacks of a "Discussion" section, where the authors underline the value of their work with reference to the
current state of the art.  Please rework accordingly and provide a better explanation of the key features of this review by adding also a reworked 
"Conclusion" section.

Author Response

Comments and Suggestions for Authors

Title: "A review on mechanical models for cellular medium: investigation on material characterization and numerical simulation”

This manuscript reviews current key mechanical models for cellular medium, trying to track their evolution from inception to the latest version. In particular, these models are categorized in terms of their mechanical modeling methods.  The authors claim that this work focuses on the importance of constitutive relationships and microstructure models in studying mechanical properties and optimizing structural design. In addition, the authors claim that the key issues and future directions are also discussed.

General comment. Unfortunately, this work should be deeply reworked, some mistakes should be corrected and its value should be better underlined in a novel version of the manuscript which perhaps will be able to meet the quality requirements of this journal.

Some detailed comments:

Table 1. Performance comparison of various cellular medium mechanical models.

*) This table is not totally clear, then it should be reworked and improved.

In addition, the description of "features" and "Weaknesses" should be more precise.

Response 1. Thank you for your suggestion, and I have revised Table 1 in the text.

Figure 3. Equipment and schematic diagram of (a) universal testing machine (quasi-static, ~100 s-1) [32], (b) Split Hopkinson Pressure Bar (SHPB, 101-105 s-1) [33], and (c) light gas gun (106 s-1~) [34]

*) The authors should improve this figure and provide high resolution figures.

Please improve also the caption.

Response 2. Thank you for your suggestion, and I have already modified Figure 3 and its caption in the text.

Table 2. Performance comparison of various cellular medium mechanical models.

*) This table with relevant information should be better presented. Please rework accordingly

Response 3. Thank you for your suggestion, and I have revised the title of Table 2 in the text.

3.1. Phenomenological constitutive model

The phenomenological constitutive model is the expression between stress and strain obtained by fitting experimental data. It can be used to describe physical phenomena, but it cannot explain the mechanism [77].

*) The authors should explain what is the meaning of "is the expression between stress and strain" as well as the meaning of "it cannot explain the mechanism".

Perhaps they should express also which is the mechanism involved.

In materials science there are several descriptions of the behavior of materials and the phenomenological equations are one of them. Nevertheless, the crucial points of these methods are totally neglected in these lines. Please rework.

Response 4. Thank you for your suggestion. "The expression between stress and strain" is the functional relationship between stress and strain, which is described by mathematical formula. This mathematical formula reflects the macroscopic properties of material. We agree with your point of view: in materials science there are several descriptions of the behavior of materials and the phenomenological equations are one of them. They also express the mechanisms involved, which is the description of the compression process of the foam. When the load is initially applied, the force on the cells is very small, within the linear elastic range (linear elastic region). Continue to apply the load, the cells will undergo irreversible deformation, and the cells will collapse (plateau region). The load continues to increase, all cells collapse, the material becomes dense, showing the properties of the dense matrix material (densification region). "It cannot explain the mechanism" means that it cannot explain the microstructure deformation and failure mechanism of the foam, such as the weak part of the material, the stress concentration, and the local deformation. And it cannot explain the influence mechanism of cell structure (cell size, cell shape, and cell distribution) on the stress-strain curve of foam.

*) Each formula presented should also describe in detail the meaning of all quantities for interested readers. Please rework all this section accordingly. Please explain also the physical meaning of all parameters.

Response 5. Thank you for your suggestion, and I have already modified in the text.

3.2. Homogenization algorithm model

*) This paragraph is not clearly presented (in particular Eqs. 3.12-3.20 need to be clearly explained in details). Please rework accordingly.

Response 6. Thank you for your suggestion, and I have already modified in the text.

Figure 6. Single cell model (a) tetradecahedron model (Kelvin model) [113,114] and (d) octahedron model [115]

*) This figure should be improved as well as its caption

Response 7. Thank you for your suggestion, and I have already modified Figure 6 and its caption in the text.

Figure 7. 2D multi-cell models: (a) hexagonal cells and circular cells [119]; (b) circle arc (c-a) model [118]; (c) bubble growth 2D model [120]

Figure 8. (a) Dynamic crushing of hexagonal cells and circular cells at 100 s-1, (b) dependences of stress enhancement and gas pressure on nominal strain [119], and (c) deformation of c-a model at different impact speeds [118]

Figure 9. 3D multi-cell models: (a) Voroni model and tetradecahedron model [33,121], (b) “Ad-hoc” models with different porosities and polydispersities [130], (c) bubble growth 3D model [120], (d) Body-centred cubic (BCC), Face-centred cubic (FCC), Kelvin's tetrakaidecahedron structure [130], and (e) random model based on ellipsoidal cells [112]

Figure 10. Deformation modes of (a) Voroni model and tetradecahedron model at 8.5 m/s [121], (b) comparison of Voroni model and sample at 30 m/s and 70 m/s, (c) Voroni model at different impact velocities and crushing strength under various impact velocities [33], (d) deformation of ad-hoc model, and (e) compressive behavior of foams (Ad-hoc, BCC, FCC, Kelvin, and samples) [130]

Figure 11. (a) Flow chart of foam geometry development based on micro-CT, (b) digital foam geometry and sectional view [34], (c) the process to obtain the foam model and cell pore volume distribution and cell size volume ratio [146], and (d) X-ray microtomographic image and geometrical model [137]

Figure 12. (a) Failure process during uniaxial compression [137], (b) deformation patterns and stress-strain curves under uniaxial compressive loading and tension loading [146], and (c) cell collapse mechanism, free surface velocity-time curves, and elastic wave propagation characteristic through a 8 mm thickness foam at 845 m/s [34]

*) All these figures should be provided in high resolution and all the captions should be reworked to explain in details all the provided images. When panels are used (as in this case) a detailed caption should be provided to the interested readers. Please, check also the correctness of names and citations. What is the meaning of "Voroni" ?????

Response 8. Thank you for your suggestion, and I have already modified these figures and their captions in the text. Voronoi, also called Thiessen Polygon, is a set of continuous polygons composed of vertical bisectors connecting two adjacent points. The distance from any point in a Voronoi to the control points constituting the polygon is less than the distance to the control points of other polygons. Each Voronoi contains only one discrete point data; the distance from the point in the Voronoi to the corresponding discrete point is the closest; the distance from the point on the edge of the Voronoi to the discrete points on both sides of the polygon is equal.

  1. Conclusions

At present, researchers have performed much work on the characterizing mechanical properties, establishing the constitutive relationship and microstructure model, and optimizing cellular medium designs. This paper introduced the mechanical models of cellular medium, classified them according to their modeling methods, tracked the historical progress of key models, and summarized main achievements in this field. We focused on the importance of constitutive relationships and microstructure models in studying mechanical properties and optimizing structural design. Although there has been decades of research on the mechanical properties of cellular medium, there is still a large number of key issues to be resolved.

  • Although the constitutive model can depict the stress strain relationship of cellular medium well, it cannot explain the microstructure deformation mechanism during stretching or compressing.
  • The phenomenological constitutive model with cell parameters (cell size and shape) and a high strain rate (greater than 100 s-1) needs to be further investigated.
  • The single cell model and multi-cell model can explain the microstructure deformation mechanism of cellular medium. However, most of the models are based on various algorithms and describe the local microstructure (RVE) of cellular medium, which is greatly affected by random factors, and the stability is not high.
  • Researchers should investigate how to build a true and effective microstructure model of cellular medium and ensure the stability of the algorithm while reducing its cost.
  • The deformation mode and failure mechanism of cellular medium under explosive loading and ultrahigh strain rates, gradient cellular material (gradient mode) structural design and performance optimization [18,57,71,150-155], and functional foam composite materials (e.g., doped metal particles, carbon nanotubes, and graphene) [21,156-159] are the keys to future research on cellular medium.
  1. Future directions

The research of cellular medium is growing rapidly and has been widely used in the fields of packaging boxes, transportation, daily necessities, industrial manufacturing, and aerospace. There will be more application in the future:

  • Expand the mechanical constitutive models of cellular medium to suitable for various complex loads, and propose more constitutive models to understand the multifunctional performance of cellular medium.
  • Novel design of cellular metamaterials with unprecedented properties and advanced manufacturing technology [160-164].
  • Multifunctional applications for potential industrial applications, such as: high-performance wave absorption characteristics [165-167], electromagnetic interference shielding [158,168-170], adsorption thermodynamics and dynamics characteristics [171], and can be used in medical care, satellite communications, electronic equipment, national defense security and other fields in the future [170,172-175]. And conduct more experiments to verify potential industrial applications.

*) This work completely lacks of a "Discussion" section, where the authors underline the value of their work with reference to the current state of the art.  Please rework accordingly and provide a better explanation of the key features of this review by adding also a reworked "Conclusion" section.

Response 9. Thank you for your suggestion, and I have already modified “Conclusion” section in the text.

Reviewer 3 Report

This paper presents a review on the mechanical properties of cellular materials. The topic is interesting and can be accepted after a substantial improvement.

1. The mechanical properties covered here are not complete. The authors need to justify the focus of this article.

2. In the context of cellular lattices, many important new developments have been neglected, such as the aspect of manufacturing irregularity (International Journal of Mechanical Sciences, 150 784 - 806), mechanical properties under vibrating condition (Mechanics of Materials, 157 103796) and active property modulation (International Journal of Solids and Structures, 208-209 31-48). A review paper should be as comprehensive as possible.

3. The authors should try to classify cellular materials from the perspective of structural configs considering different aspects.

4. The writing needs to be improved.

Author Response

Comments and Suggestions for Authors

This paper presents a review on the mechanical properties of cellular materials. The topic is interesting and can be accepted after a substantial improvement.

  1. The mechanical properties covered here are not complete. The authors need to justify the focus of this article.

Response 1. Thanks for your suggestion. The characterization of the mechanical properties of cellular material in this paper is mainly quasi-static loading and dynamic loading. In the numerical simulation, it is expressed as applying tensile load, compressive load, bending load and impact speed to obtain stress-strain curves, elastic modulus, strength, etc. This is the focus of mechanical properties in this article.

  1. In the context of cellular lattices, many important new developments have been neglected, such as the aspect of manufacturing irregularity (International Journal of Mechanical Sciences, 150 784-806), mechanical properties under vibrating condition (Mechanics of Materials, 157 103796) and active property modulation (International Journal of Solids and Structures, 208-209 31-48). A review paper should be as comprehensive as possible.

Response 2. Thank you for your suggestion, and I have added these references in section 4.2 in the text.

  1. The authors should try to classify cellular materials from the perspective of structural configs considering different aspects.

Response 3. Thanks for your suggestion, and I have added classification of cellular materials in section 2 in the text.

  1. The writing needs to be improved.

Response 4. Thank you for your suggestion, I have already modified in the text. We also embellished the whole paper through the Elsevier language editor service.

Round 2

Reviewer 2 Report

Title: "A review on mechanical models for cellular medium: investigation on material characterization and numerical simulation"

The authors only partially revised text in a word editor, where novel text additions are together to the deleted text. Unfortunately, this way to present their work is not too clear so the global new version of the text is not so understandable: some parts of the main text are still not clearly presented and should be reworked. In particular, section "3.1. Phenomenological constitutive model” should be made more clear. Indeed, as already pointed out in the previous review, all the used equations should be presented in a careful way and all the used quantities should be clearly stated and explained.

In addition, the authors should better present within a "Discussion" section, which is still lacking, the value of this work with respect to the current state of the art, while within the "Conclusion" section the novelty of this work should be resumed in a clear way.

Finally, the stile of the language is still suboptimal in several part of this text, thus the interested readers can not follow the logical flow of the work in a suitable way.

Some detailed comments:

Lines: “where K is the plateau stress, n and r are constantsindexes typically in the ranges 0n249
nde0r0.3. The constitutive equation can describe the mechanical behavior well in the 250
strain range of 0.1-0.7, but it cannot fit the linear elasticity and densification region well. 251
Sherwood [82] considered the influence of the strain rate and proposed a 252
stress-strain framework of cellular medium, as shown in the following formula: “

*) the authors should rework this part and better explain the meaning of “in the ranges 0n249
nde0r0.3.”

Lines: “where f(ε) = ∑10 i=1 Aiεn is the strain function, Ai’s are determined using a tenth-order 254
polynomial fit. M(ε, ε̇) = (ε̇⁄ε̇ 0)a+bε is the modulus function, Ai, a, and b are the param- 255
eters to be identifiedconstant, and ε̇ 0 is the an arbitrary quasi-static strain rate. “

*) The authors should be better explain what is “ ε̇ 0 is the an arbitrary quasi-static strain rate”, in which way this strain rate has been calculated and what is the range for this quantity.

Lines: “where σ0(ε) represents the nominal stress response at a constant quasi-static strain 262
ratedepends on the type of cellular medium, Tr is the glass transition temperature, . and 263
a, and b are strain rate dependency coefficients., C1 and C2 are Williams-Landel-Ferry 264
(the parameters to be identifiedWLF) function coefficients to be determined from tem- 265
perature effect experiments. 26”

*) in Equation 3.6 and 3.7 the authors should better explain the meaning of a and b, their physical meaning together with a suitable range of variation for their values.

*) Similarly, Equations from 3.8 to 3.12 are not clear and need to be explained in all details for interested readers.

Lines: “ 3.2. Homogenization algorithm model 296

Cellular medium can be considered as a composite with matrix and pores. The 297

macroscopic mechanical parameters of cellular medium are obtained through macro- 298

scopic experimental tests. A representative volume unit (RVE) connects mesoscale and 299

macroscale. On the macroscale, assuming that each point is the center of the RVE, these 300

points contain potentially heterogeneous microstructures. The macroscopic strain and 301

stress are equal to the volume average of the microstrain and microstress in all unknown 302

regions within the RVE [96-98], as shown in Figure 4(b). Under linear boundary condi- 303

tions, the macroscopic stress (?) and macroscopic strain (?̅ are equivalent to the micro- 304

scopic stress (σ) and microscopic strain (ε) in the RVE region. The volume average rela- 305

tionship of the strain field between the matrix and pores in the RVE region (? is [99]:”

*) This section is not clear. The authors should better explain what is the “Homogenization procedure”. In addition, the Equations 3.14-3.22 are not clear. Please improve this section.

Lines: “The progressive damage model describes the generation and evolution process of 346
damage factorD. D can be described as a function of true stress, or a certain failure crite- 347
rion coefficient. 348
The Matzenmiller-Lubliner-Taylor (MLT) damage model , or Hashin damage mod- 349
el, describes damage factorD as a function of the Hashin failure criterion coefficient. The 350
model assumes that each phase of material obeys Weibull distribution. The damage 351
evolution process is actually the process of strain energy release. In the process of strain 352
energy release, the material will soften, and the macroscopic manifestation is the degra- 353
dation of elastic modulus and the decrease of load-bearing capacity. The damage varia- 354
ble in damage evolution is the key issue and foundation. It directly reflects the damage 355
mechanism of the material. The damage evolution functionφ(f) includes: the step func- 356
tion damage evolution model (equation 3.2024), power function damage evolution model 357
(equation 3.2125), linear softening damage evolution model (equation 3.2226) and expo- 358
nential function damage evolution model (equation 3.2327) [99,101,103]. 35

*) This part should be improved and all the symbols used in Equations 3.23-3.27 should be stated in a clear way and explained in details.

Lines: “5. Conclusions 541

At present, researchers have performed much work on the characterizing mechani- 542

cal properties, establishing the constitutive relationship and microstructure model, and 543

optimizing cellular medium designs. Combining the characterization of static mechanical 544

properties and dynamic mechanical properties, the researchers investigated the effects of 545

cell size, matrix, strain rate, and testing methods on the mechanical properties of cellular 546

medium, and established a large number of constitutive models to describe the 547

stress-strain relationship. To investigate the microstructure deformation mechanism of 548

cellular medium, researchers combined the algorithm to establish the microstructure 549

modes (single cell models and multi-cell models), which can not only obtain the 550

stress-strain curves, but also observe the cell deformation, the weak part of the material, 551

the stress concentration and the local deformation, etc. 552

This paper introduced the mechanical models of cellular medium, classified them 553

according to their modeling methods, tracked the historical progress of key models, and 554

summarized the main achievements of the predecessors,in this field. We and focused on 555

the importance of constitutive relationships and microstructure models in studying me- 556

chanical properties and optimizing structural design. Although there has been decades of 557

research on the mechanical properties of cellular medium, there is still a large number of 558

key issues to be resolved. 559

• Although the constitutive model can depict the stress strain relationship of cellular 560

medium well, it cannot explain the microstructure deformation mechanism during 561

stretching or compressing. 562

• The phenomenological constitutive model with cell parameters (cell size and shape), 563

and a high strain rate (greater than 100 104 s-1) and explosive loading needs to be 564

further investigated.

The single cell model and multi-cell model can explain the microstructure defor- 566

mation mechanism of cellular medium. However, most of the models are based on 567

various algorithms and describe the local microstructure (RVE) of cellular medium, 568

which is greatly affected by random factors, and the stability is not high. 569

• Researchers should investigate hIt is a great challengeow to build establish a stable, 570

low-cost, true and effective microstructure model of cellular medium and ensure the 571

stability of the algorithm while reducing its cost. 572

• The deformation mode and failure mechanism of cellular medium under explosive 573

loading and ultrahigh strain rates, gradient cellular material (gradient mode) struc- 574

tural design and performance optimization [18,57,71,150153-155158], and functional 575

foam composite materials (e.g., doped metal particles, carbon nanotubes, and gra- 576

phene) [21,156159-159162] are the keys to future research on cellular medium. 577

• 578

6. Future directions 579

The research of cellular medium is growing rapidly and has been widely used in the 580

fields of packaging boxes, transportation, daily necessities, industrial manufacturing, and 581

aerospace. There will be more applications in the future: 582

• Expand the mechanical constitutive models of cellular medium to suitable for vari- 583

ous complex loads, and propose more constitutive models to understand the mul- 584

tifunctional performance of cellular medium. 585

• Novel design of cellular metamaterials with unprecedented properties and ad- 586

vanced manufacturing technology [160163-164167]. 587

• Multifunctional applications for potential industrial applications, such as: 588

high-performance wave absorption characteristics [165168-167170], electromagnetic 589

interference shielding [158161,168171-170173], adsorption thermodynamics and 590

dynamics characteristics [171174], and can be used in medical care, satellite com- 591

munications, electronic equipment, national defense security and other fields in the 592

future [170173,172175-175178]. And conduct more experiments to verify potential 593

industrial applications.”

*) The authors could see to the general comment to improve these parts. In addition, the “Conclusion” section should be the final one. The “future directions” should be inserted within the “Discussion” section.

Author Response

Response to the referees’ comments and modifications:

Reviewer 2

The authors only partially revised text in a word editor, where novel text additions are together to the deleted text. Unfortunately, this way to present their work is not too clear so the global new version of the text is not so understandable: some parts of the main text are still not clearly presented and should be reworked. In particular, section "3.1. Phenomenological constitutive model” should be made more clear. Indeed, as already pointed out in the previous review, all the used equations should be presented in a careful way and all the used quantities should be clearly stated and explained.

In addition, the authors should better present within a "Discussion" section, which is still lacking, the value of this work with respect to the current state of the art, while within the "Conclusion" section the novelty of this work should be resumed in a clear way.

Finally, the style of the language is still suboptimal in several part of this text, thus the interested readers can not follow the logical flow of the work in a suitable way.

Response:

Thanks for your suggestions. The sections "3.1. Phenomenological Constitutive Model", "5. Discussion" and "6. Conclusion" were modified in accordance with the given suggestions. And all the parameters in the equations have been explained. We have embellished the entire paper through the Elsevier language editing service.

Some detailed comments:

Comment 1:

Lines: “where K is the plateau stress, n and r are indexes typically in the ranges 0≤n≤nde0≤r≤0.3. The constitutive equation can describe the mechanical behavior well in the strain range of 0.1-0.7, but it cannot fit the linear elasticity and densification region well.

Sherwood [82] considered the influence of the strain rate and proposed a stress-strain framework of cellular medium, as shown in the following formula:

*) the authors should rework this part and better explain the meaning of “in the ranges 0≤n≤nde0≤r≤0.3.”

Response 1:

Thank you for your suggestion. “in the ranges 0≤n≤nde0≤r≤0.3.” is “in the ranges 0≤n≤4, 0≤r≤0.3.” already modified in the manuscript.

Comment 2:

Lines: “where is the strain function, Ai’s are determined using a tenth-order polynomial fit. is the modulus function. a and b are constant, and ε̇0 is an arbitrary quasi-static strain rate. “

*) The authors should be better explain what is “ε̇0 is an arbitrary quasi-static strain rate”, in which way this strain rate has been calculated and what is the range for this quantity.

Response 2:

Sherwood [82] defined "ε̇0 is an arbitrary quasi-static strain rate", which is the initial value of strain rate calculation in the range of 4.233*10-3-8.466*10-2 s-1. The initial value of the strain rate is calculated under quasi-static loading:

where is the velocity under quasi-static loading, and is the original length of the material.

Comment 3:

Lines: “where σ0(ε) represents the nominal stress response at a constant quasi-static strain rate, Tr is the glass transition temperature. a and b are strain rate dependency coefficients. C1 and C2 are Williams-Landel-Ferry (WLF) function coefficients to be determined from temperature effect experiments.”

*) in Equation 3.6 and 3.7 the authors should better explain the meaning of a and b, their physical meaning together with a suitable range of variation for their values.

Response 3:

In Equation 3.6 and 3.7, n(ε)=a+bε is a linear function of strain, which is the slope of the relationship between the stress and strain rate. a and b are material constants in the ranges of 0-0.0442 and 0.0343-0.145.

Nagy [83] defined “n is a linear function of strain (i.e., n = a+bε, a, b are constants). The functional form of n(ε) is material-dependent. Table 2 shows expressions of n(ε) for different foams.”

Zhang [84] defined “a and b are material constants. To obtain material constants, stress σ is plotted against the strain rate on the log-log scale for different strain levels. For selected foams, it is noticed that the data form a family of straight lines with tangent n that is approximately a linear function of strain ε.”

Comment 4:

*) Similarly, Equations from 3.8 to 3.12 are not clear and need to be explained in all details for interested readers.

Response 4:

In the manuscript, the Equation 3.8 was obtained by adding a modulus function to Equation 3.3. The meanings of A, B, E, m and n are reflected in equation (3.3) [80]. Jeong proposed two modulus function , as shown in equation (3.9) [85] and equation (3.10) [86].

In Equation 3.9, Jeong [85] defined “The strain rate function is based on the assumption that log-log plot of stress-strain data for a certain strain exhibits a linear function of strain. Careful examination of experimental data shows that this is not true especially for the semi-flexible urethane. A new strain rate function is introduced to describe the log-log plot of stress vs. strain rate, which is concave up, at constant strain as follows:”. And he also indicated “The evaluation of these parameters is divided into two steps. First, five parameters, E, A, B, m, and n, are determined by fitting the experimental stress-strain data at the reference strain rate (quasi-static test at 0.001 s-1 strain rate). Next, the remaining two parameters, a and b, are found by linearly approximating the experimental stress-strain data for another strain rate (0.1 s-1). Most of the parameters are considered to be dependent on the material, density, and temperature. Thus, two-step experiments should be done on the specimens with the same density at the same temperature.”

In Equation 3.10, Jeong [85] thought “The modulus function works as a scale factor of the shape function linearly varying along the strain axis between a reference strain rate (quasi-static strain rate 0.001 s-1) and another strain rate (0.1 s-1). Through several experiments, however, the slope of the scale factor of the densification region was a little different from that of the plateau region. Here we propose a new modulus function.” In Equation 3.11 and 3.12, He defined “The suggested modulus function has different expressions of n(ε) depending on three strain regions. A cubic polynomial function of ε is used on the transition zone, which consists of plateau and densification regions adjacent to the critical strain. Linear functions of n(ε) are used for plateau and densification regions. Note that the critical strain belongs to the transition zone and that ε1 has smaller value and ε2 has larger value than εcr, respectively. Four coefficients, , , and , can be obtained through boundary conditions at and . Four parameters, , , and , are found by fitting the experimental stress-strain data at another strain rate (at the strain rate of 0.1 s-1). Once four parameters, , , and , are determined, , , and can be obtained by solving the above four linear simultaneous equations.”

Comment 5:

Lines: “ 3.2. Homogenization algorithm model

Cellular medium can be considered as a composite with matrix and pores. The macroscopic mechanical parameters of cellular medium are obtained through macroscopic experimental tests. A representative volume unit (RVE) connects mesoscale and macroscale. On the macroscale, assuming that each point is the center of the RVE, these points contain potentially heterogeneous microstructures. The macroscopic strain and stress are equal to the volume average of the microstrain and microstress in all unknown regions within the RVE [96-98], as shown in Figure 4(b). Under linear boundary conditions, the macroscopic stress () and macroscopic strain ( are equivalent to the microscopic stress (σ) and microscopic strain (ε) in the RVE region. The volume average relationship of the strain field between the matrix and pores in the RVE region () is [99]:”

*) This section is not clear. The authors should better explain what is the “Homogenization procedure”. In addition, the Equations 3.14-3.22 are not clear. Please improve this section.

Response 5:

The homogenization algorithm model combines a macro-scale and a micro-scale to predict the effect of microstructure on the overall properties of the material. The two scales are connected by a representative volume element (RVE). In a micro–macro-approach, at each macro-point , we know the macro-strain and need to compute the macro-stress . At micro-level, the macro-point is viewed as the center of a RVE with domain and boundary . If linear boundary conditions (BCs) are applied to the RVE, then the problem of relating macro-strains and macro-stresses can be transformed onto that of relating average strains and average stresses over the RVE. The specific calculation process and equations 3.14-3.18 have been clearly explained in the manuscript.

The homogenization model is only available when the matrix phase and the inclusion phase are both isothermal linear elastic materials. However, the cellular medium is not just a linear elastic material. To study the plasticity of material, it is necessary to adopt a progressive damage model. The progressive damage model introduces damage variables to express the microscopic defects in the entire material. It can be expressed that the stiffness of the material with damage is lower than the stiffness of the non-damaged material. The specific calculation process and equations 3.19-3.22 have been clearly explained in the manuscript.

Comment 6:

Lines: “The progressive damage model describes the generation and evolution process of damage factor D. D can be described as a function of true stress, or a certain failure criterion coefficient.
The Matzenmiller-Lubliner-Taylor (MLT) damage model , or Hashin damage model, describes damage factor D as a function of the Hashin failure criterion coefficient. The model assumes that each phase of material obeys Weibull distribution. The damage evolution process is actually the process of strain energy release. In the process of strain energy release, the material will soften, and the macroscopic manifestation is the degradation of elastic modulus and the decrease of load-bearing capacity. The damage variable in damage evolution is the key issue and foundation. It directly reflects the damage mechanism of the material. The damage evolution function φ(f) includes: the step function damage evolution model (equation 3.24), power function damage evolution model (equation 3.25), linear softening damage evolution model (equation 3.26) and exponential function damage evolution model (equation 3.27) [99,101,103].“

*) This part should be improved and all the symbols used in Equations 3.23-3.27 should be stated in a clear way and explained in details.

Response 6:

The progressive damage model describes the generation and evolution of damage variable D. Damage evolution is a function of plasticity: damage initiates at the onset of plasticity. The damage variable D in damage evolution is the key issue and foundation. It directly reflects the damage mechanism of the material. In damage mechanics, damage variable D is used to describe the degradation process of material. Generally, in the non-destructive state, and in the complete damage state. The Hashin damage model describes damage factor D as a function of the Hashin failure criterion coefficient f. The functional relationship between damage factor D and failure factor f can be defined as a variety of functional forms, so that the occurrence and evolution of damage can be described flexibly. The damage evolution function includes: the step function damage evolution model, power function damage evolution model, linear softening damage evolution model and exponential function damage evolution model. The symbols used in Equations 3.23-3.27 have been clearly explained in the manuscript.

Comment 7:

Lines: “5. Conclusions

At present, researchers have performed much work on the characterizing mechanical properties, establishing the constitutive relationship and microstructure model, and optimizing cellular medium designs. Combining the characterization of static mechanical properties and dynamic mechanical properties, the researchers investigated the effects of cell size, matrix, strain rate, and testing methods on the mechanical properties of cellular medium, and established a large number of constitutive models to describe the stress-strain relationship. To investigate the microstructure deformation mechanism of cellular medium, researchers combined the algorithm to establish the microstructure modes (single cell models and multi-cell models), which can not only obtain the stress-strain curves, but also observe the cell deformation, the weak part of the material, the stress concentration and the local deformation, etc.

This paper summarized the main achievements of the predecessors, and focused on the importance of constitutive relationships and microstructure models in studying mechanical properties and optimizing structural design. Although there has been decades of research on the mechanical properties of cellular medium, there is still a large number of key issues to be resolved.

  • Although the constitutive model can depict the stress strain relationship of cellular medium well, it cannot explain the microstructure deformation mechanism during stretching or compressing.
  • The phenomenological constitutive model with cell parameters (cell size and shape), and a high strain rate (greater than 104 s-1) and explosive loading needs to be further investigated.
  • The single cell model and multi-cell model can explain the microstructure deformation mechanism of cellular medium. However, most of the models are based on various algorithms and describe the local microstructure (RVE) of cellular medium, which is greatly affected by random factors, and the stability is not high.
  • It is a great challenge to establish a stable, low-cost, true and effective microstructure model of cellular medium.
  • The deformation mode and failure mechanism of cellular medium under explosive loading and ultrahigh strain rates, gradient cellular material (gradient mode) structural design and performance optimization [18,57,71, 153-158], and functional foam composite materials (e.g., doped metal particles, carbon nanotubes, and graphene) [21, 159-162] are the keys to future research on cellular medium.
  1. Future directions

The research of cellular medium is growing rapidly and has been widely used in the fields of packaging boxes, transportation, daily necessities, industrial manufacturing, and aerospace. There will be more applications in the future:

  • Expand the mechanical constitutive models of cellular medium to suitable for various complex loads, and propose more constitutive models to understand the multifunctional performance of cellular medium.
  • Novel design of cellular metamaterials with unprecedented properties and advanced manufacturing technology [163-167].
  • Multifunctional applications for potential industrial applications, such as: high-performance wave absorption characteristics [168-170], electromagnetic interference shielding [161, 171-173], adsorption thermodynamics and dynamics characteristics [174], and can be used in medical care, satellite communications, electronic equipment, national defense security and other fields in the future [173, 175-178]. And conduct more experiments to verify potential industrial applications.”

*) The authors could see to the general comment to improve these parts. In addition, the “Conclusion” section should be the final one. The “future directions” should be inserted within the “Discussion” section.

Response 7:

Thank you for your suggestion. We added the “Discussion” section, and changed the position of “Conclusions” section in the manuscript according to your suggestion as follows:

  1. Discussion

At present, researchers have performed much work in characterizing mechanical properties, establishing constitutive relationship and microstructure model, and optimizing cellular medium designs. This paper reviewed the mechanical properties characterization of cellular medium, and provided the latest comments of key mechanical models, including classifying these models based on modeling scale and methodology, tracking the historical progress of these models, and discussing their scope of application, advantages and disadvantages.

  • The effects of cell size, matrix, relative density, strain rate and testing methods on the mechanical properties of cellular medium were systematically investigated through quasi-static mechanical testing and dynamic mechanical testing. The investigation on mechanical properties of cellular medium under explosive loading and ultrahigh strain rates involves complex stress waves, and systematic analysis methods and influence laws have not yet been obtained.
  • The constitutive model can quickly and accurately describe the stress-strain relationship of the cellular medium, and obtain mechanical parameters (elastic modulus, plateau stress, densification strain, etc.). But it cannot explain the microstructure deformation mechanism during stretching or compression. The constitutive model with cell parameters (cell size and cell shape), high strain rate (greater than 104 s-1) and explosive loading needs further study.
  • The microstructure model can not only obtain the stress-strain curve, but also explain the microstructure deformation mechanism of the cellular medium, such as cell deformation, weak parts of the material, stress concentration and local deformation. However, most of the models are based on various algorithms and describe the local microstructure (RVE) of cellular medium, which is greatly affected by random factors and is not stable. It is a great challenge to establish a stable, low-cost, real and effective microstructure model of cellular medium.
  • Molecular dynamics, phase field method and peridynamics can be used to establish nanoscopic models to investigate the structural evolution and properties of molecules and atoms in cellular medium. Due to these methods involve nanoscale, simulation results and experimental characterization results are not comparable, and the scale range is small and the calculation amount is large, so they are not introduced in this paper.

The research of cellular medium is growing rapidly and has been widely used in the fields of packaging boxes, transportation, daily necessities, industrial manufacturing, and aerospace. There will be more applications in the future:

  • The deformation mode and failure mechanism of cellular medium under explosive loading and ultrahigh strain rates, the structural design and performance optimization of gradient cellular material [18,57,71, 153-158], and functional foam composite materials (e.g., doped metal particles, carbon nanotubes, and graphene) [21, 159-162] are the keys to future research on cellular medium.
  • Expand the mechanical constitutive models of cellular medium to suitable for various complex loads, and propose more constitutive models to understand the multifunctional performance of cellular medium.
  • Novel design of cellular metamaterials with unprecedented properties and advanced manufacturing technology [163-167].
  • Molecular dynamics [169-171], phase field method [172-174], peridynamics [175] combined with multi-physics coupling calculations can obtain the nanoscale structural evolution and multifunctional performance of cellular medium.
  • Multifunctional applications for potential industrial applications, such as: high-performance wave absorption characteristics [176-178], electromagnetic interference shielding [162,179-181], adsorption thermodynamics and dynamics characteristics [182], and can be used in medical care, satellite communications, electronic equipment, national defense security and other fields in the future [181,183-186]. And conduct more experiments to verify potential industrial applications.

  1. Conclusions

This paper mainly focuses on the research progress of cellular medium materials in recent years, introduces the mechanical properties characterization of cellular medium, and provides mechanical models to describe the stress-strain relationship. The phenomenological constitutive models and the homogenization algorithm models can quickly obtain the stress-strain relationship of cellular medium, but they cannot investigate the mechanism of the microstructure deformation. The microstructure models can not only obtain the stress-strain curve, but also observe the microstructure deformation of the cellular medium, but these models are not stable and require a large amount of calculation. The molecular dynamics, phase field method, peridynamics are the research focus of cellular medium at the molecular scale and atomic scale. This paper provides a reference and a compilation of important documents for researchers in the same research direction, and points out the research focus and application directions.

Reviewer 3 Report

accept

Author Response

Thanks a lot!